# Approximate Gradient Coding for Distributed Learning with Heterogeneous Stragglers

**Heekang Song**
Korea Advanced Institute of Science and Technology
School of Electrical Engineering
hghsong@kaist.ac.kr

**Wan Choi**
Seoul National University
Department of Electrical and Computer Engineering
wanchoi@snu.ac.kr

## Abstract

In this paper, we propose an optimally structured gradient coding scheme to mitigate the straggler problem in distributed learning. Conventional gradient coding methods often assume homogeneous straggler models or rely on excessive data replication, limiting performance in real-world heterogeneous systems. To address these limitations, we formulate an optimization problem minimizing residual error while ensuring unbiased gradient estimation by explicitly considering individual straggler probabilities. We derive closed-form solutions for optimal encoding and decoding coefficients via Lagrangian duality and convex optimization, and propose data allocation strategies that reduce both redundancy and computation load. We also analyze convergence behavior for $\lambda$-strongly convex and $\mu$-smooth loss functions. Numerical results show that our approach significantly reduces the impact of stragglers and accelerates convergence compared to existing methods.

## 1 Introduction

In recent years, the rapid advancements in deep learning have underscored the significance of large datasets and large-scale AI models as critical components for performance enhancement. Breakthrough models such as ChatGPT, Gemini, and SORA have not only demonstrated unprecedented capabilities but have also transformed the industrial landscape, reshaping how AI technologies are applied across various domains. These models are commonly trained using gradient-descent-based algorithms, but the training process for such large-scale models demands immense computational resources, such as GPUs and NPUs, leading to a *computation bottleneck*. To address these challenges, building on foundational frameworks like MapReduce [1] and Spark [2], distributed computing [3, 4] has emerged as a promising and practical solution, and evolving into distributed learning, which mitigates computation and communication bottlenecks in large-scale training scenarios.

A distributed learning architecture typically consists of a central coordinator (master node) that trains an AI model using gradient-based optimization (e.g., Gradient Descent (GD)) to minimize a given loss function, partitions and distributes the dataset to worker nodes for parallel computation of local gradients, and aggregates these local gradients to approximate the global gradient and update model parameters. However, heterogeneity in computational and communication resources can cause bottlenecks, particularly due to the slowest worker node, known as a *straggler*, impairing overall efficiency and scalability.

39th Conference on Neural Information Processing Systems (NeurIPS 2025).

Motivated by this aspect, *Gradient Coding*, a specialized technique for mitigating stragglers in distributed learning, has been extensively studied [5–12]. The concept was first introduced in [5], where data replication enables coding of partial gradients. Assuming a known number of stragglers, it designed linear encoding and decoding schemes and identified fundamental data replication limits. Its extension [6] explored trade-offs between communication overhead and resilience to stragglers, proposing gradient coding schemes that reduce overhead at the expense of lower resilience. However, these methods assume prior knowledge of straggler counts—often unrealistic in practice—and require extensive data replication, placing a heavy computational burden on worker nodes.

To overcome limitations associated with unrealistic straggler models and high computational costs, *Approximate Gradient Coding* has emerged as a practical solution under probabilistic straggler models, explored in prior works [7–12]. Unlike exact gradient coding [5, 6], which precisely reconstructs the true gradient sum, approximate gradient coding relies on estimated gradient sum for model updates. This relaxation is justified since optimization methods, such as Stochastic Gradient Descent (SGD), naturally tolerate approximations and noise, still ensuring convergence. By alleviating the computational burden of exact recovery, approximate gradient coding improves efficiency and mitigates straggler impacts through minimizing the residual error between true and approximate gradient sum.

The authors of [7] introduced the approximate gradient coding framework by leveraging the normalized adjacency matrix of an expander graph to construct encoding and decoding schemes. Similarly, [8] proposed an approximate gradient coding scheme based on sparse random graphs, where gradient components are assigned to worker nodes via Bernoulli sampling and the residual error is controlled through probabilistic guarantees. The study [9] further analyzed the residual error of Fractional Repetition (FR) codes, originally introduced in [5], noting that these codes can only be constructed when the number of distributed nodes is a multiple of the data replication factor. Additionally, [10] examined the fundamental trade-off among the data replication factor, the number of stragglers, and the residual error in approximate gradient coding. Building on these advancements, the authors of [11] proposed a novel approximate gradient coding scheme that leverages expander graphs while dynamically optimizing decoding coefficients to minimize residual error.

While previous studies [7–11] primarily aimed to reduce residual errors between the gradient estimator and the true gradient, this focus alone does not necessarily ensure model convergence. Therefore, robust convergence guarantees become essential when using approximate gradient sums in gradient-descent algorithms. Stochastic Gradient Coding (SGC) [12] addressed this by introducing a pairwise data distribution scheme, where the number of worker nodes sharing any two data partitions is proportional to the product of their respective replication factors. SGC designs its encoding coefficients to ensure an unbiased gradient estimator using binary decoding and provides a thorough convergence analysis. Empirical results showed that SGC performs robustly even under severe straggler conditions, where residual-error-focused methods [7–11] may falter. Unlike exact gradient coding—which requires each data partition to be replicated more than the number of stragglers—SGC enforces constraints on pairwise data distribution. Consequently, depending on the total number of data partitions, their replication factors, and the number of workers, it can become challenging or even infeasible to satisfy the pairwise distribution constraints.

In summary, previous studies have generally pursued two directions. The first focuses on minimizing the residual error but often lacks rigorous convergence analysis or relies on binary encoding/decoding coefficients for analytical simplicity. The second ensures the gradient estimator's unbiasedness by carefully designing encoding coefficients while using binary decoding coefficients. Observing the progress in both directions, we believe that integrating these approaches can potentially yield improved performance. Moreover, most prior research assumes homogeneous straggler scenarios with uniform straggling probabilities across worker nodes, which is unrealistic given the varying computation and communication capacities encountered in practical settings. Under a non-uniform probabilistic model, gradient updates may consistently neglect certain datasets, increasing generalization errors and the risk of converging to local optima. To address these issues, we propose a novel approximate gradient coding technique specifically designed for heterogeneous straggler environments.

## 2 Preliminaries

### 2.1 Distributed Learning with Gradient Coding

Consider a distributed learning, where a master node aims to solve an optimization problem using a gradient-descent-like algorithm across $k$ worker nodes. Given a dataset $\mathcal{D} = \{\mathcal{D}_i\}_{i=1}^n$ consisting of $n$ data partitions, the goal is to learn a parameter $\boldsymbol{\beta}^* \in \mathbb{R}^l$ that minimizes a loss function $L(\mathcal{D}, \boldsymbol{\beta})$. This process involves iteratively solving:

$$\boldsymbol{\beta}^* = \arg\min_{\boldsymbol{\beta}} L(\mathcal{D}, \boldsymbol{\beta}), \tag{1}$$

by approximating the parameter update:

$$\boldsymbol{\beta}_{t+1} = \boldsymbol{\beta}_t - \gamma_t \cdot g^{(t)}, \tag{2}$$

where $g^{(t)}$ represents the aggregated gradient at iteration $t$,

$$g^{(t)} = \nabla L(\mathcal{D}, \boldsymbol{\beta}_t) = \sum_{i=1}^n \nabla L(\mathcal{D}_i, \boldsymbol{\beta}_t), \tag{3}$$

and $\gamma_t$ is a learning rate. In a distributed framework, the master node divides the dataset $\mathcal{D}$ into $k$ batches $(\mathcal{B}_1, \mathcal{B}_2, \ldots, \mathcal{B}_k)$ and distributes each data batch $\mathcal{B}_j$ to its corresponding worker node $j$, where data batches may overlap and do not necessarily have the same size. Each worker node $j$ then computes the local partial gradient $\nabla L(\mathcal{B}_j, \boldsymbol{\beta}_t)$ in parallel, and then the master node aggregates these to approximate the global gradient. Stragglers, however, can impede this process; gradient coding combats them by injecting redundancy and applying coding across batches to tolerate slow or failed worker nodes.

The gradient coding procedure consists of three phases—data distribution, local computation, and gradient update phase. In *data distribution phase*, the master node replicates each partition $\mathcal{D}_i$ to $d_i$ worker nodes to tolerate stragglers. This redundancy enables worker nodes to send encoded partial gradients (via linear combinations), so the master node can recover the true gradient sum even if some worker nodes straggle. This process introduces the parameter $d = \frac{1}{n}\sum_{i=1}^n d_i$, known as the *computation load* (or *replication factor*), which quantifies the average redundancy in computation (or data replication). Hereinafter, we will refer to the replication factor as the computation load.

Subsequently, in *local computation phase*, each worker node $i$ computes the partial gradient from its assigned data batch $\mathcal{B}_i$, i.e., $\{g_j^{(t)} : \forall \mathcal{D}_j \in \mathcal{B}_i\}$. Using the computed partial gradients, each worker node generates an encoded message, which is then sent to the master node:

$$f_i(\boldsymbol{\beta}_t) = \sum_{\mathcal{D}_j \in \mathcal{B}_i} a_{i,j} \cdot g_j^{(t)}, \tag{4}$$

where $a_{i,j} \in \mathbb{R}$ represents the encoding coefficient used by worker node $i$ for the gradient $g_j^{(t)} = \nabla L(\mathcal{D}_j, \boldsymbol{\beta}_t)$, which corresponds to data partition $\mathcal{D}_j$.

Then, in *gradient update phase*, the master node aggregates all the received responses from non-straggling worker nodes with the decoding coefficients:

$$\hat{g}^{(t)} = \sum_{i=1}^k \mathbb{I}_i \cdot w_i \cdot f_i(\boldsymbol{\beta}_t), \tag{5}$$

where $w_i \in \mathbb{R}$ is the decoding coefficient for worker node $i$ and $\mathbb{I}_i$ is the indicator function for worker node $i$ being non-straggler, i.e.,

$$\mathbb{I}_i = \begin{cases} 1, & \text{if worker node } i \text{ is non-straggler,} \\ 0, & \text{otherwise.} \end{cases} \tag{6}$$

Unlike the homogeneous straggler assumptions in [7–13], we adopt a heterogeneous model: each worker node $i$ independently straggles at each iteration with probability $p_i$, so $\mathbb{E}[\mathbb{I}_i] = 1 - p_i$. This reflects real-world variability in computational and communication capacities. The parameter update follows the process outlined in (2); however, instead of the true gradient sum $g^{(t)}$, an estimated gradient sum $\hat{g}^{(t)}$ from gradient decoding is used for the update. Once updated, the parameters $\boldsymbol{\beta}_{t+1}$ are distributed to the worker nodes. Once the data distribution is done, the local computation and gradient-update phases repeat every iteration until convergence.

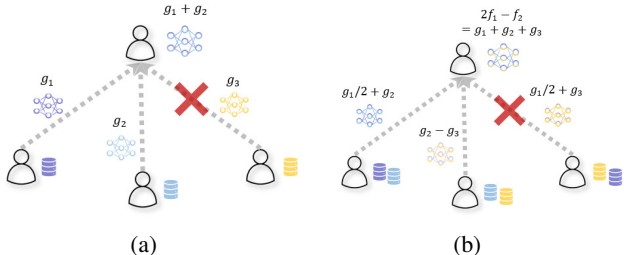

Figure 1: Motivating example of gradient coding.

## 2.2 Motivations

Figure 1 illustrates our motivating example. Without replication or coding, a single straggler forces the master node to update using only the remaining gradients—e.g., in Figure 1(a), $\boldsymbol{\beta}$ is updated with $g_1 + g_2$. By contrast, in Figure 1(b) each partition is assigned to two worker nodes, which compute and linearly encode their gradients into $f_1 = g_1/2 + g_2$, $f_2 = g_2 - g_3$, and $f_3 = g_1/2 + g_3$. Even if the third worker node becomes a straggler, the master node still recovers the true gradient sum $g = g_1 + g_2 + g_3$ by computing $2f_1 - f_2$. This perfect recovery nullifies the impact of stragglers, ensuring seamless gradient-descent updates.

From this perspective, exact gradient coding provides a systematic way to choose the encoding coefficients $a_{i,j}$ and the decoding coefficients $w_i$ so that each worker node encodes its gradients into $f_i$, and the master node recovers the true gradient sum from non-straggling messages. Representing the encoding matrix as $A$ and decoding vector as $w$, the goal is to satisfy $Aw = \mathbb{I}$, ensuring $\hat{g}^{(t)} = g^{(t)}$. Moreover, to tolerate up to $s$ stragglers, the computation load $d$ must satisfy $d \geq s + 1$, which represents a fundamental limit.

To address the high computation load and the impractical requirement of knowing the exact number of stragglers, approximate gradient coding was developed based on the homogeneous probabilistic straggler model ($p_1 = p_2 = \cdots = p_k$). Unlike exact recovery under deterministic straggler scenarios illustrated in Figure 1(b), approximate gradient coding focuses on designing the encoding matrix $A$ and decoding vector $w$ to minimize the average residual error between the true gradient sum $g^{(t)}$ and its estimated counterpart $\hat{g}^{(t)}$. Prior studies have demonstrated that smaller residual errors and higher computation load typically improve convergence behavior. Nevertheless, as highlighted in [12], exclusively minimizing residual error may lead to degraded convergence performance in environments with frequent straggling events. In contrast, ensuring an unbiased gradient estimator provides robust convergence properties even under severe straggler conditions.

## 3 Optimally Structured Gradient Coding

Building on these insights, our gradient coding approach aims not only to reduce both the overall residual error and the variance of the estimator but also to leverage the unbiasedness of gradient estimators in heterogeneous straggler environments.[1] By addressing these factors, the proposed method enhances convergence performance, achieving faster convergence with stronger theoretical guarantees. Accordingly, the encoding matrix $A$ and decoding vector $w$, each comprising the encoding coefficients $\{a_{i,j}, \forall i, j\}$ and decoding coefficients $\{w_i, \forall i\}$, are designed by optimizing the problem:

$$(\textbf{P1}) \quad \underset{A,w}{\text{minimize}} \quad \mathbb{E}_t[\|g^{(t)} - \hat{g}^{(t)}\|_2^2] \tag{7}$$

$$\text{subject to} \quad \mathbb{E}_t[\hat{g}^{(t)}] = g^{(t)}, \tag{8}$$

where $\mathbb{E}_t[\cdot] \triangleq \mathbb{E}[\cdot | \boldsymbol{\beta}_t]$ represents the expectation over the random behavior of the stragglers in the $t$-th iteration, conditioned on the model parameter $\boldsymbol{\beta}_t$. Since straggler effects are i.i.d. across iterations, the optimized encoding and decoding coefficients remain fixed, eliminating the need to re-optimize (**P1**) at every iteration—unless the underlying straggler statistics change.

---

[1]With an unbiased gradient estimator, the residual error equals its variance.

However, problem (**P1**) is not directly solvable, as it requires knowledge of the true gradient values, necessitating the dynamic design of gradient codes at each iteration—an impractical requirement for constructing gradient codes.

To address this challenge, we leverage a mild assumption of the true gradient:

**Assumption 1.** *(Boundedness of gradient) There exists a constant $C$ such that*

$$\|\nabla L(\mathcal{D}_j, \boldsymbol{\beta}_t)\|_2^2 = \|g_j^{(t)}\|_2^2 \le C, \forall j \in [1:n]. \tag{9}$$

Note that this approach remains practical, as it accommodates constraints imposed by activation functions or gradient clipping techniques, as outlined in [12–14]. Accordingly, we introduce the following lemma.

**Lemma 1.** *Suppose that **Assumption 1** is satisfied and gradient estimator $\hat{g}^{(t)}$ is unbiased. Then,*

$$\mathbb{E}_t[\|g^{(t)} - \hat{g}^{(t)}\|_2^2] \le C\left[\sum_{i=1}^k p_i(1 - p_i) \cdot w_i^2 \left(\sum_{j=1}^n a_{i,j}\right)^2\right]. \tag{10}$$

The proof is provided in Appendix F.1.

Since the objective is for the gradient estimator—affected by straggler behavior of worker nodes—to mimic the target gradient computed by gradient-based algorithms at each iteration, based on the given data partitions $\mathcal{D}$ and model parameter $\boldsymbol{\beta}_t$, the source of randomness lies in the straggler behavior at each iteration. Consequently, we have

$$\mathbb{E}_t[\hat{g}^{(t)}] = \sum_{j=1}^n g_j^{(t)} \cdot \mathbb{E}_t\left[\sum_{i=1}^k \mathbb{I}_i \cdot w_i a_{i,j}\right]. \tag{11}$$

To ensure that this expected value matches the true gradient sum $g^{(t)} = \sum_{j=1}^n g_j^{(t)}$ regardless of the specific values of $g_j^{(t)}$, the following unbiasedness condition must be satisfied individually:

$$\mathbb{E}_t\left[\sum_{i=1}^k \mathbb{I}_i \cdot w_i a_{i,j}\right] = \sum_{i=1}^k (1 - p_i) \cdot w_i a_{i,j} = 1, \ \forall j \in [1:n], \tag{12}$$

where the equality comes from the fact that $\mathbb{E}_t[\mathbb{I}_i] = 1 - p_i$.

From these observations, the original problem (**P1**) can be reformulated as:

$$(\mathbf{P2}) \quad \underset{A,w}{\text{minimize}} \quad \sum_{i=1}^k \delta_i \tilde{w}_i^2 \left(\sum_{j=1}^n a_{i,j}\right)^2 \tag{13}$$

$$\text{subject to} \quad \sum_{i=1}^k \tilde{w}_i a_{i,j} = 1, \ \forall j \in [1:n], \tag{14}$$

where $\tilde{w}_i = (1 - p_i) \cdot w_i$ and $\delta_i = p_i/(1 - p_i)$. It is apparent that the optimization problem (**P2**) is non-convex, primarily due to the strong coupling between the encoding and decoding variables. However, both the objective function and the constraints exhibit a similar structure with respect to the combined variables $\tilde{w}_i a_{i,j}$. By defining $\alpha_i^j = \tilde{w}_i a_{i,j}$, (**P2**) can, without loss of optimality, be transformed into

$$(\mathbf{P3}) \quad \underset{\boldsymbol{\alpha}}{\text{minimize}} \quad \sum_{i=1}^k \delta_i \left(\sum_{j=1}^n \alpha_i^j\right)^2 \tag{15}$$

$$\text{subject to} \quad \sum_{i=1}^k \alpha_i^j = 1, \ \forall j \in [1:n], \tag{16}$$

where $\boldsymbol{\alpha}$ is the matrix whose $(i, j)$ element is $\alpha_i^j$. Problem (**P3**) is a convex problem with respect to the transformed variables $\alpha_i^j$ and can be solved by the standard convex optimization tool, such as CVX [15] and YALMIP [16].

Subsequently, for a given $\boldsymbol{\alpha}$, the encoding and decoding coefficient can be obtained with randomly generated $\tilde{w}_i$ ($\tilde{w}_i \neq 0$) as:

$$a_{i,j} = \frac{\alpha_i^j}{\tilde{w}_i} \text{ and } w_i = \frac{\tilde{w}_i}{1 - p_i}, \ \forall i, j. \tag{17}$$

Note that the entry $\alpha_i^j$ is relevant to the encoding and decoding coefficient of the partial gradient computed from partition $\mathcal{D}_j$ that is attributed to worker node $i$. This representation facilitates efficient management of data distribution and redundancy across multiple worker nodes within the gradient coding framework: specifically, $\alpha_i^j \neq 0$ means data partition $\mathcal{D}_j$ is allocated to worker node $i$. Thus, the structure of matrix $\boldsymbol{\alpha}$ explicitly encodes how data partitions are distributed among worker nodes.

## 3.1 Optimal Structure of Gradient Coding

While problem (**P3**) can be solved using convex optimization tools, we also provide an opportunity to uncover the optimal structure of gradient codes that minimize the residual error while satisfying the gradient estimator's unbiasedness. This structure facilitates the development of a closed-form solution for gradient code design. To begin, we present the optimal structure of gradient codes that minimizes (**P3**).

**Theorem 1.** *The optimal structure of optimization problem (**P3**) satisfies the conditions below:*

$$\sum_{j=1}^{n} \alpha_i^j = Y_i, \forall i \in [1:k], \text{ and } \sum_{i=1}^{k} \alpha_i^j = 1, \forall j \in [1:n], \tag{18}$$

*where $Y_i = \delta_i^{-1} \cdot \frac{n}{\sum_{j=1}^{k} \delta_j^{-1}}$ and $\delta_i^{-1} = \frac{1 - p_i}{p_i}$ for all $i \in [1:k]$.*

The proof is provided in Appendix F.2. Note that the row-wise sum constraint, $\sum_{i=1}^{k} \alpha_i^j = 1$ for all $j \in [1:n]$, ensures the unbiasedness of the gradient estimator. The values of the matrix $\boldsymbol{\alpha}$ that satisfy the optimal structure of gradient coding are determined by $Y_i$, which, in turn, depends on the straggler probabilities of all worker nodes. As a result, the gradient code is directly influenced by the straggling characteristics of the distributed nodes.

According to Theorem 1, any gradient code satisfying the optimal structure adheres to the following lemma:

**Lemma 2.** *For any gradient codes satisfying the optimal structure described in Theorem 1, the residual error of gradient estimator is bounded by*

$$\mathbb{E}_t[\|g^{(t)} - \hat{g}^{(t)}\|_2^2] \leq n^2 C \cdot \frac{1}{\sum_{i=1}^{k} \delta_i^{-1}} \tag{19}$$

*and squared norm of the gradient estimator is bounded by*

$$\mathbb{E}_t[\|\hat{g}^{(t)}\|_2^2] \leq n^2 C \cdot \left(1 + \frac{1}{\sum_{i=1}^{k} \delta_i^{-1}}\right). \tag{20}$$

The proof is provided in Appendix F.3.

## 3.2 Optimally Structured Gradient Code Construction

There may be multiple configurations of $\boldsymbol{\alpha}$ that can adhere to the optimal structure outlined in Theorem 1. In this subsection, we detail two closed-form configurations-termed *Scheme I* and *Scheme II*—that not only satisfy the optimal structure but also reduce the computation load.

Throughout, without loss of generality, we assume that $p_1 \leq p_2 \leq \cdots \leq p_k$. To effectively capture more gradient information across the dataset on average, it is advantageous to allocate more data to worker nodes with a lower likelihood of becoming stragglers. Accordingly, a gradient coding strategy can be designed so that worker nodes with lower indices are assigned a proportionally larger share of the data. Let $b_1 \geq b_2 \geq \cdots \geq b_k$ denote the number of partitions assigned to worker nodes 1 through $k$, selected from $n$ distinct and non-overlapping data partitions.[2] For both *Scheme I* and *Scheme II*,

---

[2]The data distribution parameters $b_1, ..., b_{k-1}$ can be adjusted based on the straggler probabilities and the available storage (or computing) capacity.

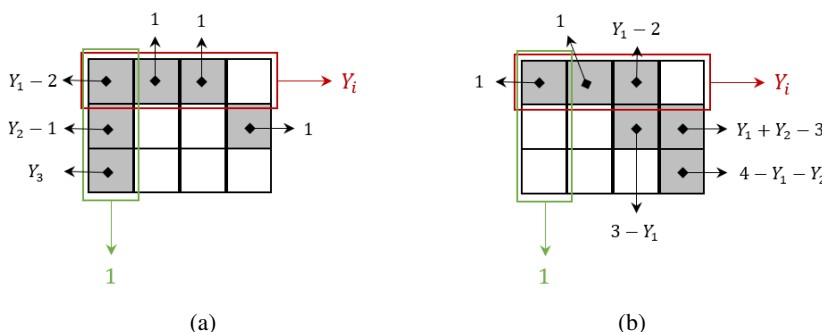

Figure 2: Illustrative example of the proposed schemes: (a) Scheme I and (b) Scheme II.

the values $b_i$ are chosen to satisfy $\sum_{i=1}^{k} b_i = n + k - 1$, which ensures that the $k$ batches collectively cover all $n$ partitions, with each adjacent pair of worker nodes sharing exactly one partition. This design reduces the overall computation load on individual worker nodes.

**Scheme I**  A single, specific data partition ($\mathcal{D}_1$) is a common partition shared by all workers, while the remaining partitions are assigned exclusively to individual workers. This scheme has a centralized sharing structure. The data allocation is performed as:

- *Worker node 1* is assigned the first $b_1$ data partitions: $\mathcal{B}_1 = \{\mathcal{D}_1, \mathcal{D}_2, \ldots, \mathcal{D}_{b_1}\}$.
- *Worker node $i \in [2 : k-1]$* is assigned a batch that includes the shared data partition $\mathcal{D}_1$, along with $b_i - 1$ additional data partitions that are exclusive to worker node $i$ and not shared with any other worker node. Specifically, for $2 \leq i \leq k - 1$, $\mathcal{B}_i = \{\mathcal{D}_1\} \cup \{\mathcal{D}_{l'_i}, \mathcal{D}_{l'_i+1}, \ldots, \mathcal{D}_{l'_i+b_i-2}\}$, where $l'_1 = 2$, and for $i \geq 2$, the starting index is recursively defined as $l'_i = l'_{i-1} + b_{i-1} - 1$, ensuring that each worker node receives the next $b_i - 1$ unassigned, non-overlapping data partitions.
- *Worker node $k$* is assigned only the shared data partition: $\mathcal{B}_k = \{\mathcal{D}_1\}$.

Given the data distribution in *Scheme I*, we set the elements of $\boldsymbol{\alpha}$ to meet the optimal structure, which can be divided into three cases:

- If a partition $\mathcal{D}_j$ is not distributed to worker node $i$ (i.e., $\mathcal{D}_j \notin \mathcal{B}_i$), we set $\alpha_i^j = 0$.
- If a partition $\mathcal{D}_j$ is distributed exclusively to worker node $i$ (i.e., $\mathcal{D}_j \in \mathcal{B}_i$ but $\mathcal{D}_j \notin \mathcal{B}_\ell$ for any $\ell \neq i$), we set $\alpha_i^j = 1$, which satisfies the unbiasedness condition.
- If a partition $\mathcal{D}_j$ is distributed to all worker nodes, i.e., $\mathcal{D}_j = \mathcal{D}_1$, $\alpha_i^j$ is set to be $Y_i - b_i + 1$, ensuring that Theorem 1 is satisfied.

*Example (Figure 2(a)):*  Consider $n = 4$ data partitions and $k = 3$ worker nodes. One feasible choice of $\{b_i\}$ is $b_1 = 3$, $b_2 = 2$, $b_3 = 1$. The batches in *Scheme I* become $\mathcal{B}_1 = \{\mathcal{D}_1, \mathcal{D}_2, \mathcal{D}_3\}$, $\mathcal{B}_2 = \{\mathcal{D}_1, \mathcal{D}_4\}$, and $\mathcal{B}_3 = \{\mathcal{D}_1\}$. Here all worker nodes share $\mathcal{D}_1$, worker node 1 exclusively has $D_2, D_3$, and worker node 2 exclusively has $D_4$. The $\boldsymbol{\alpha}$ assignments for this example are:

$$\begin{pmatrix} \alpha_1^1 & \alpha_1^2 & \alpha_1^3 & \alpha_1^4 \\ \alpha_2^1 & \alpha_2^2 & \alpha_2^3 & \alpha_2^4 \\ \alpha_3^1 & \alpha_3^2 & \alpha_3^3 & \alpha_3^4 \end{pmatrix} = \begin{pmatrix} Y_1 - 2 & 1 & 1 & 0 \\ Y_2 - 1 & 0 & 0 & 1 \\ Y_3 & 0 & 0 & 0 \end{pmatrix}, \tag{21}$$

where each column sum is 1 (for $D_1$, $\alpha_1^1 + \alpha_2^1 + \alpha_3^1 = Y_1 + Y_2 + Y_3 - 3 = 1$; for $D_2, D_3, D_4$, the sum is trivially 1 since each is only held by one worker node), and each row $i$ sums to $Y_i$.

**Scheme II**  Each worker shares exactly one data partition with the worker of the adjacent index, so every partition is held by at most two workers. This scheme has a sequential and decentralized sharing structure. The data distribution is as follows:

- *Worker node 1* is assigned the first $b_1$ data partitions: $\mathcal{B}_1 = \{\mathcal{D}_1, \mathcal{D}_2, \ldots, \mathcal{D}_{b_1}\}$.

- *Worker node* $i \in [2 : k-1]$ is assigned $b_i$ consecutive data partitions, starting from the last partition of the previous worker node's batch to ensure exactly one sharing partition between adjacent worker nodes. Specifically, we define $\mathcal{B}_i = \{\mathcal{D}_{l_i}, \mathcal{D}_{l_i+1}, \ldots, \mathcal{D}_{l_i+b_i-1}\}$, where $l_1 = 1$ and $l_i = l_{i-1} + b_{i-1} - 1$ for $i \geq 2$. By this construction, each batch $\mathcal{B}_i$ shares exactly one partition with the preceding batch $\mathcal{B}_{i-1}$ (specifically, partition $\mathcal{D}_{l_i} = \mathcal{D}_{l_{i-1}+b_{i-1}-1}$). Consequently, each data partition $\mathcal{D}_j$ resides on at most two worker nodes, being either exclusive to one worker node or shared between two consecutive worker nodes.

- *Worker node* $k$ is assigned the last $b_k = 1$ data partition. i.e., $\mathcal{B}_k = \{\mathcal{D}_n\}$.

Given the data distribution in *Scheme II*, we determine the elements of $\boldsymbol{\alpha}$ to satisfy the optimal structure specified in Theorem 1, which can be categorized into the three cases:

- If a partition $\mathcal{D}_j$ is not distributed to worker node $i$ (i.e., $\mathcal{D}_j \notin \mathcal{B}_i$), we set $\alpha_i^j = 0$.

- If a partition $\mathcal{D}_j$ is distributed exclusively to worker node $i$ (i.e., $\mathcal{D}_j \in \mathcal{B}_i$ but $\mathcal{D}_j \notin \mathcal{B}_\ell$ for any $\ell \neq i$), we set $\alpha_i^j = 1$, which maintains the unbiasedness condition.

- If a partition $\mathcal{D}_j$ is shared between worker node $i$ and worker node $i+1$ (i.e., $\mathcal{D}_j = \mathcal{B}_i \cap \mathcal{B}_{i+1}$), $\alpha_i^j$ and $\alpha_{i+1}^j$ are carefully assigned to ensure that $\alpha_i^j + \alpha_{i+1}^j = 1$, $\sum_{j=1}^n \alpha_i^j = Y_i$, and $\sum_{j=1}^n \alpha_{i+1}^j = Y_{i+1}$ from the conditions in Theorem 1. To construct such a matrix $\boldsymbol{\alpha}$, we can recursively determine the nonzero elements row by row. Starting from worker node 1, we assign the first $b_1 - 1$ elements in the row as 1 (corresponding to exclusive partitions), and set the last element $\alpha_1^{b_1} = Y_1 - \sum_{q=1}^{b_1-1} \alpha_1^q$ to satisfy the row-wise sum constraint, $\sum_{j=1}^n \alpha_i^j = Y_i$. Then, since partition $\mathcal{D}_{b_1}$ is shared with worker node 2, the corresponding entry is set as $\alpha_2^{l_2} = \alpha_2^{b_1} = 1 - \alpha_1^{b_1}$. The same procedure is repeated recursively: for each worker node $i$, once $\alpha_i^{l_i}$ is determined by the previous row, the remaining values in the row are set to satisfy the row-wise sum constraint $\sum_{j=1}^n \alpha_{i,j} = Y_i$. This recursive process ensures that all entries of $\boldsymbol{\alpha}$ satisfy both the optimal row-sum and column-sum conditions.

*Example (Figure 2(b)):* Consider $n = 4$ and $k = 3$ with $b_1 = 3$, $b_2 = 2$, $b_3 = 1$. This yields the batches $\mathcal{B}_1 = \{\mathcal{D}_1, \mathcal{D}_2, \mathcal{D}_3\}$, $\mathcal{B}_2 = \{\mathcal{D}_3, \mathcal{D}_4\}$, and $\mathcal{B}_3 = \{\mathcal{D}_4\}$. Here, $\mathcal{D}_3$ is stored by both worker node 1 and worker node 2, and $\mathcal{D}_4$ is stored by both worker node 2 and worker node 3. The corresponding matrix $\alpha$ can be constructed as:

$$\begin{pmatrix} \alpha_1^1 & \alpha_1^2 & \alpha_1^3 & \alpha_1^4 \\ \alpha_2^1 & \alpha_2^2 & \alpha_2^3 & \alpha_2^4 \\ \alpha_3^1 & \alpha_3^2 & \alpha_3^3 & \alpha_3^4 \end{pmatrix} = \begin{pmatrix} 1 & 1 & Y_1 - 2 & 0 \\ 0 & 0 & 3 - Y_1 & Y_1 + Y_2 - 3 \\ 0 & 0 & 0 & 4 - Y_1 - Y_2 \end{pmatrix}, \tag{22}$$

where $4 - Y_1 - Y_2$ equals to $Y_3$ (since $\sum_{i=1}^k Y_i = \sum_{i=1}^k \delta_i^{-1} \cdot \frac{n}{\sum_{j=1}^k \delta_j^{-1}} = n$). It is evident that the sum of each row $i$ equals $Y_i$ and the sum of each column is 1, respectively, thus confirming that this construction adheres to the optimal structure outlined in Theorem 1.

The closed-form expressions of both schemes are provided in Appendix A.

**Code construction and computation load**    Based on the matrix $\boldsymbol{\alpha}$, we construct the encoding and decoding coefficient by using (17). For both *Schemes I* and *II*, the computation load $d$ remains strictly less than 2, i.e., $d < 2$. This indicates that robust performance is achieved without placing an excessive computational burden on the worker nodes. The efficiency of this code design is underscored by the fact that the computation load is $d = 1 + \frac{k-1}{n}$. Since the number of computing nodes is generally less than or equal to the size of the dataset (i.e., $k \leq n$), the resulting computation load $d$ remains below 2. This low value reflects efficient resource utilization, minimizing redundant computation while maintaining a balanced workload across the nodes.

**Theoretical analysis of convergence behavior**    Due to the page limit, the theoretical convergence analysis of our proposed method is provided in Appendix B.

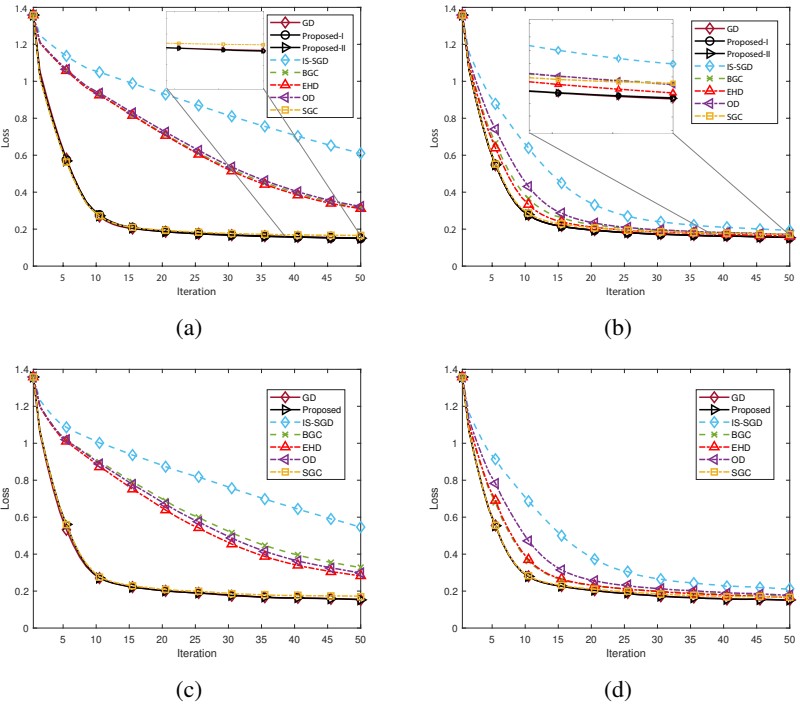

(a)

(b)

(c)

(d)

Figure 3: Convergence graph with respect to the training iteration $T$: (a) $\tau_{th} = 1.1$ ($k = 10$) (b) $\tau_{th} = 1.5$ ($k = 10$) (c) $\tau_{th} = 1.1$ ($k = 100$) (d) $\tau_{th} = 1.5$ ($k = 100$).

## 4 Experiments

In this section, we demonstrate the effectiveness of the proposed optimally structured gradient coding scheme for straggler mitigation in distributed learning. We numerically evaluate its performance on the large-scale COCO dataset [21]. In our experiments, we employ the MobileNetV3 model, and the learning rate is set to $\gamma_t = 0.01$. Suppose $\tau_{th}$ denote the response time limit for each training iteration. A worker node $i \in [1 : k]$ is classified as a straggler if its overall delay $\tau_i$ for local gradient computation and communication exceeds this limit, i.e., $\tau_i > \tau_{th}$. Then, the straggler probability of each worker node $i \in [1 : k]$ can be modeled by

$$p_i = e^{-\psi_i(\tau_{th}-1)}, \tag{23}$$

where $\psi_i$ represents the straggling parameter of the worker node $i$ and $\tau_{th} \geq 1$ [18, 19]. In these experiments, the straggling parameter is sampled from the uniform distribution [19], i.e., $\psi_i \sim \text{Uniform}(\psi_{\min}, \psi_{\max})$. We set $k = 10, \psi_{\min} = 0.1, \psi_{\max} = 2$, and $\tau_{th} = 1.1$, unless stated otherwise. The experimental results are obtained by averaging the outcomes of 10 simulation runs.

We compare our design with centralized learning-based GD, Ignore-Stragglers SGD (IS-SGD), Bernoulli Gradient Coding (BGC) [8], ERASUREHEAD (EHD) [9], Optimal Decoding (OD) [11], Stochastic Gradient Coding (SGC) [12]. The implementation details are in Appendix D.

Figure 3 shows the model convergence as a function of the training iteration $T$ and the per-iteration response time limit $\tau_{th}$. Throughout the experiments, the loss represents the overall training objective of the object-detection model—the sum of classification and bounding-box regression losses—computed on the COCO validation set. The bounding-box term is computed as a Smooth-L1 regression on the predicted center offsets and log-scale width/height adjustments with respect to each ground-truth box. In Figure 3(a), we illustrate the convergence behavior for $k = 10$ and $\tau_{th} = 1.1$. Except for SGC—which guarantees unbiasedness of the gradient estimator—all benchmark methods suffer from poor convergence due to the adverse impact of stragglers. In particular, for IS-SGD, when straggler tendencies are high, the learning speed can be severely compromised, underscoring the necessity of gradient coding techniques for straggler mitigation in distributed learning. Moreover, merely ensuring unbiasedness is not sufficient for optimal performance; further reducing the estimator's residual

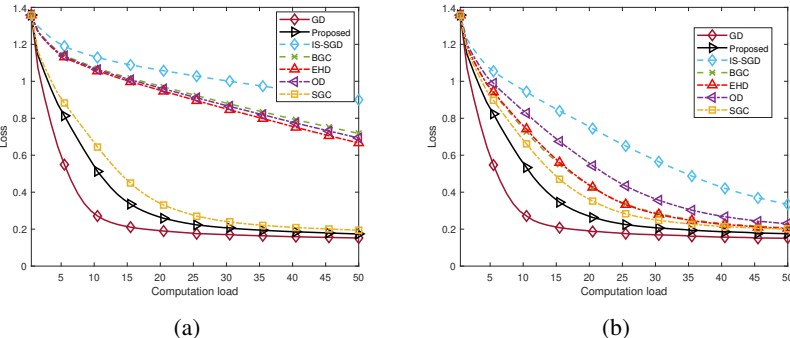

Figure 4: Convergence graph with respect to the computation load $d$: (a) $\tau_{th} = 1.1$ (b) $\tau_{th} = 1.5$.

error (and variance) leads to additional improvements. Both the proposed schemes, Scheme I and Scheme II (referred to as Proposed-I and Proposed-II), detailed in Section 3.2, not only adhere to the optimal structure but also demonstrate rapid convergence. They closely emulate the behavior of GD in environments unaffected by straggler effects. In particular, Figure 3(b) presents the convergence for $k = 10$ and $\tau_{th} = 1.5$. As the per-iteration response time limit $\tau_{th}$ increases, the likelihood of straggling decreases (due to the straggler model in (23)), thereby accelerating the convergence of the benchmark methods. Nevertheless, our proposed schemes still achieve a faster convergence rate, closely matching that of GD. Additionally, Figs. 3(c) and 3(d) show the convergence for a larger distributed system with $k = 100$, for $\tau_{th} = 1.1$ and $\tau_{th} = 1.5$, respectively. These results demonstrate that our proposed approach effectively mitigates the impact of stragglers regardless of the number of distributed nodes, even with a lower computation load (i.e., $d = 1 + \frac{k-1}{n} < 2$) compared to the benchmarks.

In Figure 4, we plot the model convergence with respect to the computation load $d$ when $\tau_{th} = 1.1$ and $\tau_{th} = 1.5$. These results normalize convergence behavior by computation load, emphasizing the computational requirements necessary for each gradient coding method to achieve convergence. The simulation result consistently demonstrates that our proposed method surpasses benchmark performances, achieving rapid convergence with reduced computational effort.

Although centralized learning-based GD can achieve rapid convergence with the same level of computation load, largely because it is unaffected by stragglers, it necessitates processing the entire dataset sequentially on a single device, which considerably extends the per-iteration runtime. In contrast, its distributed version, IS-SGD, which partitions the dataset into disjoint batches and assigns them to different worker nodes without redundancy, experiences substantial performance degradation due to the adverse impact of stragglers in distributed learning environments. Consequently, our proposed method not only reduces the time per iteration relative to centralized learning but also effectively mitigates the detrimental effects of stragglers without incurring excessive computational overhead in distributed settings, thereby delivering superior performance compared to traditional benchmarks.

## 5 Conclusions

We proposed an optimally structured gradient coding scheme for mitigating stragglers in heterogeneous distributed learning systems. By deriving optimal encoding and decoding coefficients, our method minimizes residual error while maintaining an unbiased gradient estimator. Theoretical analysis and simulations showed that it consistently outperforms benchmarks, achieving high efficiency with low computation load, even in computation-limited scenarios. These results highlight its practicality for real-world distributed learning with rapid, redundancy-efficient updates.

## 6 Acknowledgements

This work was supported by the National Research Foundation of Korea(NRF) grant funded by the Korea government (MSIT) (RS-2021-NR059011).

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

# A   Closed-form Expressions of the Proposed Schemes

## A.1   Scheme I

We obtain the following closed-form expression for the elements of $\boldsymbol{\alpha}$ that meet the conditions specified in Theorem 1: For all $i \in [1 : k]$,

$$\alpha_i^j = \begin{cases} Y_i - b_i + 1, & j = 1, \\ 1, & j \in [l_i' : l_i' + b_i - 2], \end{cases} \tag{24}$$

Otherwise, for all $i \in [1 : k]$ and $\forall \mathcal{D}_j \notin \mathcal{B}_i$, $\alpha_i^j = 0$.

## A.2   Scheme II

We derive the closed-form expression for the elements of $\boldsymbol{\alpha}$ that satisfy Theorem 1: For $i = 1$,

$$\alpha_1^j = \begin{cases} 1, & j \in [1 : b_1 - 1], \\ Y_1 - b_1 + 1, & j = b_1. \end{cases} \tag{25}$$

For $i \in [2 : k - 1]$,

$$\alpha_i^j = \begin{cases} 1 - \alpha_{i-1}^j, & j = l_i, \\ 1, & j \in [l_i + 1 : l_i + b_i - 2], \\ Y_i + \alpha_{i-1}^j - b_i + 1, & j = l_i + b_i - 1. \end{cases} \tag{26}$$

For $i = k$,

$$\alpha_k^n = Y_k. \tag{27}$$

Otherwise, for all $i \in [1 : k]$ and $\forall \mathcal{D}_j \notin \mathcal{B}_i$, $\alpha_i^j = 0$.

# B Theoretical Analysis

In this section, we conduct a convergence analysis of our optimally structured gradient coding scheme across a variety of loss functions. To facilitate the derivation, we impose certain assumptions on the loss function as:

**Assumption 2.** *($\lambda$-strongly convexity) The loss function $L$ is $\lambda$-strongly convex if for all $\boldsymbol{\beta}, \boldsymbol{\beta}' \in \mathbb{R}^l$,*

$$L(\boldsymbol{\beta}) \geq L(\boldsymbol{\beta}') + \langle \nabla L(\boldsymbol{\beta}'), \boldsymbol{\beta} - \boldsymbol{\beta}' \rangle + \frac{\lambda}{2} \left\| \boldsymbol{\beta} - \boldsymbol{\beta}' \right\|_2^2. \tag{28}$$

**Assumption 3.** *($\mu$-smoothness) The loss function $L$ is $\mu$-smooth if for all $\boldsymbol{\beta}, \boldsymbol{\beta}' \in \mathbb{R}^l$, $\mu \geq 0$,*

$$L(\boldsymbol{\beta}) \leq L(\boldsymbol{\beta}') + \langle \nabla L(\boldsymbol{\beta}'), \boldsymbol{\beta} - \boldsymbol{\beta}' \rangle + \frac{\mu}{2} \left\| \boldsymbol{\beta} - \boldsymbol{\beta}' \right\|_2^2. \tag{29}$$

Here, $\langle \cdot, \cdot \rangle$ denotes the inner product operation. Note that **Assumptions 2** and **3** regarding the loss function are commonly satisfied by a wide range of standard learning models, such as logistic regression and softmax classifiers.

We first derive the convergence proof for a $\lambda$-strongly convex loss function, as presented in the following theorem.

**Theorem 2.** *Suppose the loss function $L$ satisfies **Assumptions 1** and **2**. Then, by setting $\gamma_t = 1/(\lambda t)$, it holds for any optimally structured gradient codes that*

$$\mathbb{E}[\|\boldsymbol{\beta}_T - \boldsymbol{\beta}^*\|_2^2] \leq \frac{4n^2 C}{\lambda^2 T} \left( 1 + \frac{1}{\sum_{i=1}^k \delta_i^{-1}} \right), \tag{30}$$

*where $\delta_i^{-1} = \frac{1-p_i}{p_i}$.*

The proof is provided in Appendix F.4. Theorem 2 shows that with a strongly convex loss and a decaying learning rate $\gamma_t = \frac{1}{\lambda t}$, the proposed method achieves a convergence rate of $O(1/T)$, comparable to classical SGD. Notably, the error bound reflects straggler heterogeneity through the scaling term $\left( 1 + \frac{1}{\sum_{i=1}^k \delta_i^{-1}} \right)$, ensuring robust convergence even with non-uniform straggling. Importantly, convergence depends only on straggler probabilities, not on computation load, meaning even minimal data replication suffices. This contrasts with SGC [12], where convergence relies on the minimum computation load $\min_{i \in [1:k]} d_i$, highlighting the stronger robustness of our approach.

Next, we extend our convergence analysis to $\mu$-smooth loss functions to address non-convex settings. Based on the **Assumption 3**, we establish the following theorem.

**Theorem 3.** *Suppose the loss function $L$ satisfies **Assumptions 1** and **3**. Then, it holds for any optimally structured gradient codes:*

*By setting $\gamma_t = \gamma = 1/(T+1)^{1/2}$,*

$$\frac{1}{T+1} \sum_{t=0}^{T} \mathbb{E}[\|g^{(t)}\|_2^2] \leq \frac{L(\boldsymbol{\beta}_0) - L(\boldsymbol{\beta}^*)}{(T+1)^{1/2}} + \frac{1}{(T+1)^{1/2}} \frac{\mu n^2 C}{2} \cdot \left( 1 + \frac{1}{\sum_{i=1}^k \delta_i^{-1}} \right), \tag{31}$$

*where the following limit holds:*

$$\lim_{T \to \infty} \frac{L(\boldsymbol{\beta}_0) - L(\boldsymbol{\beta}^*) + \frac{\mu n^2 C}{2} \cdot \left( 1 + \frac{1}{\sum_{i=1}^k \delta_i^{-1}} \right)}{(T+1)^{1/2}} = 0. \tag{32}$$

*By setting $\gamma_t = 1/(t+1)^{1/2}$,*

$$\frac{1}{T+1} \sum_{t=0}^{T} \mathbb{E}[\|g^{(t)}\|_2^2] \leq \frac{L(\boldsymbol{\beta}_0) - L(\boldsymbol{\beta}^*)}{(T+1)^{1/2}} + \frac{\mu n^2 C(1 + \log(T+1)^{1/2}) \left( 1 + \frac{1}{\sum_{i=1}^k \delta_i^{-1}} \right)}{(T+1)^{1/2}}, \tag{33}$$

*where the following limits hold:*

$$\lim_{T \to \infty} \frac{L(\boldsymbol{\beta}_0) - L(\boldsymbol{\beta}^*)}{(T+1)^{1/2}} + \frac{\mu n^2 C(1 + \log(T+1)^{1/2}) \left( 1 + \frac{1}{\sum_{i=1}^k \delta_i^{-1}} \right)}{(T+1)^{1/2}} = 0. \tag{34}$$

The proof is provided in Appendix F.5. Theorem 3 shows that for $\mu$-smooth loss functions, the proposed gradient coding scheme guarantees convergence to a stationary point. Specifically, whether a constant learning rate $\gamma_t = \frac{1}{(T+1)^{1/2}}$ or a decaying learning rate $\gamma_t = \frac{1}{(t+1)^{1/2}}$ is employed, the average squared gradient norm approaches zero as the number of iterations increases. This result confirms that the algorithm remains effective even in non-convex settings.

Lastly, we conduct an extended analysis for loss functions that satisfy both $\lambda$-strong convexity and $\mu$-smoothness, as detailed below.

**Theorem 4.** *Suppose the loss function $L$ satisfies **Assumptions 1, 2** and **3**. Then, it holds for any optimally structured gradient codes that*

$$\mathbb{E}[\|\boldsymbol{\beta}_{t+1} - \boldsymbol{\beta}^*\|_2^2] \leq \|\boldsymbol{\beta}_0 - \boldsymbol{\beta}^*\|_2^2 \cdot \prod_{p=0}^{t}(1 - \gamma_p \lambda) + n^2 C \cdot \left(1 + \frac{1}{\sum_{i=1}^{k} \delta_i^{-1}}\right) \cdot \sum_{p=0}^{t} \gamma_p^2 \prod_{q=p+1}^{t}(1 - \gamma_q \lambda). \quad (35)$$

*By setting $\gamma_t = \gamma < 1/\lambda$,*

$$\lim_{t \to \infty} \mathbb{E}[\|\boldsymbol{\beta}_{t+1} - \boldsymbol{\beta}^*\|_2^2] < \frac{n^2 C}{\lambda^2} \cdot \left(1 + \frac{1}{\sum_{i=1}^{k} \delta_i^{-1}}\right). \quad (36)$$

*By setting $\gamma_t = 1/(\lambda t)$,*

$$\lim_{t \to \infty} \mathbb{E}[\|\boldsymbol{\beta}_{t+1} - \boldsymbol{\beta}^*\|_2^2] \leq 0. \quad (37)$$

The proof is provided in Appendix F.6. Theorem 4 shows that under optimally structured gradient coding in a heterogeneous straggler setting, the model error comprises two parts: one diminishing with the learning rate and strong convexity, and another reflecting accumulated variance from stragglers. With a constant learning rate ($\gamma_t = \gamma < 1/\lambda$), the algorithm quickly converges to a bounded neighborhood of the optimum, where the neighborhood size depends on straggler variability. In contrast, a decaying rate ($\gamma_t = 1/(\lambda t)$) ensures convergence to the true optimum over time. This highlights a key trade-off: constant rates yield faster initial progress, while decaying rates achieve asymptotic optimality.

# C Discussions

## C.1 Practical Example of Probabilistic Straggler Modeling

Suppose $\tau_{th}$ denote the response time limit for each training iteration. A worker node $i \in [1:k]$ is classified as a straggler if its overall delay $\tau_i$ for local gradient computation and communication exceeds this limit, i.e., $\tau_i > \tau_{th}$, and the straggler probability of worker node $i$ can be obtained by

$$p_i = Pr(\tau_i > \tau_{th}). \tag{38}$$

The modeling of stragglers depends on the system's primary bottleneck—whether it lies in computation or communication. For instance, in communication-limited environments, straggler behavior in wireless networks is often modeled by an exponential distribution [17]. In contrast, for computation-limited settings, computing latency is typically characterized by a shifted-exponential distribution [18, 19].

Intuitively, a worker node's likelihood of becoming a straggler is influenced not only by its inherent stochastic latency—driven by factors such as computation and communication capacity—but also by the response time threshold, $\tau_{th}$. A tighter threshold increases the probability of stragglers but accelerates each training iteration. This reveals a fundamental trade-off between mitigating straggler effects and achieving faster model updates. Our experiments empirically demonstrate this trade-off and show that the proposed scheme achieves an effective balance between the two.

## C.2 Worst-Case Computation Load

The maximum computation load on a single worker is a critical metric for practical deployment, and our framework is explicitly designed to address this. In our schemes, each worker $i$'s computation load is $b_i$, the number of assigned data partitions. Thus, the worst-case computation load is $\max_{i \in [1:k]} b_i$.

Our method allows for flexible—though not entirely arbitrary—adjustment of the parameters $b_1, \ldots, b_k$ based on straggler probabilities and the computational capacity of each worker. The proposed Schemes I and II leverage the constraint $\sum_{i \in [1:k]} b_i = n + k - 1$ to reduce the computation load, which inherently imposes a mathematical upper bound on the worst-case computation load, given by $\max_{i \in [1:k]} b_i \leq n - k + 2$. For instance, this bound can be achieved by assigning $b_1 = n - k + 2$, $b_2 = 2, \ldots, b_{k-1} = 2$, and $b_k = 1$. Moreover, assuming the total dataset can be partitioned into subsets of approximately equal size, the number of data partitions $n$ can also be selected accordingly. Under this scenario, choosing $n = k$ ensures that the worst-case computation load is at most 2.

Thus, system designers can flexibly constrain $\max_{i \in [1:k]} b_i$ based on specific hardware limitations, effectively preventing any single worker from becoming overloaded while still optimizing overall system performance. For example, the example (Figure 2) in Section 3.2 presents a scenario with $n = 4$ data partitions and $k = 3$ worker nodes, where the allocation parameters are set as $b_1 = 3$, $b_2 = 2$, and $b_3 = 1$. In this case, the worst-case computation load is $\max_{i \in [1:k]} b_i = 3$. To reduce this worst-case load, the parameters can be rebalanced to $b_1 = 2$, $b_2 = 2$, and $b_3 = 2$, resulting in a lower worst-case computation load of $\max_{i \in [1:k]} b_i = 2$.

Importantly, the average computation load $d = (n + k - 1)/n$ remains strictly below 2 (since $k \leq n$), ensuring overall efficiency. In summary, our framework offers both per-worker load control and high average efficiency, making it well-suited to heterogeneous real-world systems.

## C.3 Sparsity of Optimally Structured Gradient Coding

The number of non-zero entries in the matrix $\boldsymbol{\alpha}$ corresponds to those in the encoding matrix $A$, which directly determines the computation load on each distributed node. Therefore, constructing a sparse $\boldsymbol{\alpha}$ matrix is equivalent to reducing the computation load.

In this context, we focus on constructing a sparse matrix. The matrix $\boldsymbol{\alpha}$ can be represented as a bipartite graph $G = (R, C, E)$, where the row and column indices of $\boldsymbol{\alpha}$ form two disjoint vertex sets, $R$ and $C$, respectively. An edge $(i, j)$ exists between $i \in R$ and $j \in C$ if and only if $\alpha_i^j \neq 0$, with the corresponding edge weight given by $\alpha_i^j$. To preserve the optimal structure of the gradient coding scheme, the graph must satisfy two constraints: for each vertex $i \in R$, the sum of the weights of

edges incident to $i$ must equal $Y_i$; and for each vertex $j \in C$, the sum of the weights of edges incident to $j$ must equal 1.

Assume that $G$ contains $o$ disjoint and connected subgraphs, denoted as $S_1, S_2, \ldots, S_o \subseteq G$. Each subgraph $S_l$ consists of a row vertex set $R_l \subseteq R$ and a column vertex set $C_l \subseteq C$, respectively. According to the optimal structure, for each subgraph $S_l$, $\forall l \in [1:o]$, the sum of edge weights incident to each vertex $i \in R_l$ must equal $Y_i$, and the sum of edge weights incident to each vertex $j \in C_l$ must equal 1. Therefore, the total edge weight in any connected subgraph is $\sum_{i \in R_l} Y_i$, which is equal to $\sum_{j \in C_l} 1$. Thus, we have

$$\sum_{i \in R_l} Y_i = \sum_{i \in C_l} 1 = |C_l|, \forall l \in [1:o], \tag{39}$$

where this structural dependency arises because the row-wise sum and column-wise sum are combinatorially coupled. Note that the graph $G$ always satisfies the connectivity rule, i.e., $\sum_{i \in R} Y_i = |C|$, since $\sum_{i \in R} Y_i = \sum_{i=1}^{k} Y_i = \sum_{i=1}^{k} \delta_i^{-1} \cdot \frac{n}{\sum_{j=1}^{k} \delta_j^{-1}} = n$ and $|C| = n$. It is well known that any connected graph $S_l$, for $l \in [1:o]$, must contain at least $|R_l| + |C_l| - 1$ edges. In Section 3.2, we introduced a gradient code construction method for the case in which the overall graph is connected and contains exactly $n + k - 1$ edges—the minimum required for connectivity. This approach can naturally be extended to connected subgraphs. Specifically, for each connected subgraph, a submatrix is constructed using only the row indices in $R_l$ and column indices in $C_l$, following the same construction strategy. Therefore, to maximize the sparsity of the optimally structured gradient code, the overall graph $G$ should be partitioned into the largest possible number of connected subgraphs, each containing the minimum number of edges required for connectivity.

Based on these observations, we introduce a sparse code construction algorithm. Inspired by the binary tree structure, the algorithm recursively partitions the graph into two disjoint subgraphs, ensuring that the sum of edge weights in each subgraph remains an integer to satisfy the connectivity constraints. Equivalently, this can be viewed as dividing the values $Y_1, \ldots, Y_k$ into two groups such that the sum within each group is a positive integer.

Initially, we define a working set of subgroups, denoted $\mathcal{Y}$, and initialize it as $\mathcal{Y} \leftarrow \{Y_1, \ldots, Y_k\}$. At each iteration, every subgroup $\mathcal{Y}_l \in \mathcal{Y}$ is split into two disjoint subgroups, $\mathcal{Y}_l^{(1)}$ and $\mathcal{Y}_l^{(2)}$, such that $\sum_{Y_i \in \mathcal{Y}_l^{(j)}} Y_i \in \mathbb{Z}_+$ for all $j \in 1, 2$. These two subgroups then replace $\mathcal{Y}_l$ in the set $\mathcal{Y}$.

This recursive partitioning process continues until no further valid subdivisions are possible. At this point, for each remaining subgroup $\mathcal{Y}_l \in \mathcal{Y}$, we define the corresponding row vertex set as $R_l = \{i | Y_i \in \mathcal{Y}_l\}$. Starting with $l = 1$, we construct the corresponding column vertex set $C_l$ by sequentially selecting $\sum_{i \in R_l} Y_i$ elements from the global column vertex set $C$. Finally, the submatrix of $\boldsymbol{\alpha}$ corresponding to each subgraph $S_l$ is constructed using the method described in Section 3.2. Further details can be found in **Algorithm 1**.

If the maximum number of subgraphs is $o_{\mathsf{max}}$, the computation load is computed as

$$d = \frac{\sum_{l \in [1:o_{\mathsf{max}}]}(|R_l| + |C_l| - 1)}{n}. \tag{40}$$

## C.4   Extensions

### C.4.1   Extension to Mini-Batch SGD

Our method fundamentally builds upon and analyzes GD-like algorithms, but it can readily be extended to mini-batch SGD algorithms. In distributed learning, each distributed node $i$ can apply our proposed method by computing gradients using mini-batch sampling on their local data partitions $(\mathcal{D}_j, \forall j \in \mathcal{B}_i)$. This approach can effectively reduce the update time per training iteration.

However, batch sampling introduces randomness at each iteration. Hence, the expectation must now represent expectations over both the randomness from stragglers and mini-batch sampling at the $t$-th iteration, conditioned on the model parameter. Since the behaviors of stragglers and mini-batch sampling are independent, the expectation $\mathbb{E}_t[\cdot]$ now represents the expectation over both. The objective from problem (**P1**) is to minimize the residual error of the aggregated stochastic gradient

**Algorithm 1** The Sparse Code Construction Algorithm

---

Calculate $Y_1, Y_2, ..., Y_k$.

$\mathcal{Y} \leftarrow \{Y_1, Y_2, ..., Y_k\}$ and $\mathcal{Y}' = \emptyset$.

**while** $|\mathcal{Y}'| \neq |\mathcal{Y}|$ **do**

    $\mathcal{Y}' \leftarrow \mathcal{Y}$ and $\mathcal{Y} = \emptyset$.

    **for** $\mathcal{Y}_l \in \mathcal{Y}'$ **do**

        Search the two disjoint groups $\mathcal{Y}_l^{(1)}, \mathcal{Y}_l^{(2)}$ from $\mathcal{Y}_l$, satisfying $\sum_{Y_i \in \mathcal{Y}_l^{(j)}} Y_i \in \mathbb{Z}_+, \forall j \in \{1, 2\}$.

        $\mathcal{Y} \leftarrow \mathcal{Y}_l^{(1)}, \mathcal{Y}_l^{(2)}$.

    **end for**

**end while**

**for** $\mathcal{Y}_l \in \mathcal{Y}$ **do**

    Construct the subgraph $S_l$ with $R_l = \{i \mid Y_i \in \mathcal{Y}_l\}$ and $C_l = \{j \mid j \in [1 + \sum_{q=1}^{l-1} \sum_{i \in R_q} Y_i :$ $\sum_{q=1}^{l} \sum_{i \in R_q} Y_i\}$

    Construct the submatrix of $\boldsymbol{\alpha}$ corresponding to the subgraph $S_l$ as in Section 3.2.

**end for**

---

estimator, $\bar{g}^{(t)} = \sum_{i,j} \mathbb{I}_i w_i a_{i,j} \tilde{g}_j^{(t)}$, where $\tilde{g}_j^{(t)}$ is the stochastic gradient computed from a mini-batch within data partition $\mathcal{D}_j$. The objective function is thus to minimize $\mathbb{E}_t \left[ \left\| g^{(t)} - \bar{g}^{(t)} \right\|_2^2 \right]$. The residual error can be decomposed as:

$$\mathbb{E}_t \left[ \left\| g^{(t)} - \bar{g}^{(t)} \right\|_2^2 \right] = \mathbb{E}_t \left[ \left\| (g^{(t)} - \hat{g}^{(t)}) + (\hat{g}^{(t)} - \bar{g}^{(t)}) \right\|_2^2 \right]$$

$$= \underbrace{\mathbb{E}_t \left[ \left\| g^{(t)} - \hat{g}^{(t)} \right\|_2^2 \right]}_{\text{(I) Straggler Variance}} + \underbrace{\mathbb{E}_t \left[ \left\| \hat{g}^{(t)} - \bar{g}^{(t)} \right\|_2^2 \right]}_{\text{(II) Sampling Variance}}$$

This separation is valid because the cross-term is zero, as $\mathbb{E}_{\text{sample}}[\hat{g}^{(t)} - \bar{g}^{(t)}] = 0$ (due to the unbiasedness).

First, Term (I) is the original error from the paper, which is bounded as:

$$\mathbb{E}_t \left[ \left\| g^{(t)} - \hat{g}^{(t)} \right\|_2^2 \right] \leq C \sum_{i=1}^{k} \delta_i \left( \sum_{j=1}^{n} \alpha_i^j \right)^2$$

Second, for Term (II), we first take the expectation over the independent mini-batch samples:

$$\mathbb{E}_t \left[ \left\| \hat{g}^{(t)} - \bar{g}^{(t)} \right\|_2^2 \right] = \mathbb{E}_t \left[ \left\| \sum_{i,j} \mathbb{I}_i w_i a_{i,j} (g_j^{(t)} - \tilde{g}_j^{(t)}) \right\|_2^2 \right]$$

$$= \mathbb{E}_{\text{straggler}} \left[ \sum_{j=1}^{n} \mathbb{E}_{\text{sample}} \left[ \left\| g_j^{(t)} - \tilde{g}_j^{(t)} \right\|_2^2 \right] \left( \sum_{i=1}^{k} \mathbb{I}_i w_i a_{i,j} \right)^2 \right]$$

$$\leq \sigma^2 \sum_{j=1}^{n} \mathbb{E}_{\text{straggler}} \left[ \left( \sum_{i=1}^{k} \mathbb{I}_i w_i a_{i,j} \right)^2 \right],$$

where $\sigma^2$ denotes an upper bound on the variance of the mini-batch gradients, that is, $\mathbb{E}_{\text{sample}}[\|g_j^{(t)} - \tilde{g}_j^{(t)}\|_2^2] \leq \sigma^2$. Using $\mathbb{E}[X^2] = \text{Var}(X) + (\mathbb{E}[X])^2$ and the unbiasedness condition $\sum_i (1-p_i) w_i a_{i,j} =$

$\sum_i \alpha_i^j = 1$, the inner expectation becomes:

$$\mathbb{E}_{\text{straggler}}\left[\left(\sum_{i=1}^{k} \mathbb{I}_i w_i a_{i,j}\right)^2\right] = \text{Var}\left(\sum_i \mathbb{I}_i w_i a_{i,j}\right) + \left(\mathbb{E}\left[\sum_i \mathbb{I}_i w_i a_{i,j}\right]\right)^2$$

$$= \sum_{i=1}^{k} (\alpha_i^j)^2 \frac{p_i}{1-p_i} + 1^2 = 1 + \sum_{i=1}^{k} \delta_i (\alpha_i^j)^2$$

Thus, Term (II) is bounded by $\sigma^2 \sum_{j=1}^{n}\left(1 + \sum_{i=1}^{k} \delta_i (\alpha_i^j)^2\right) = \sigma^2 n + \sigma^2 \sum_{i=1}^{k} \delta_i \sum_{j=1}^{n} (\alpha_i^j)^2$.

Combining the bounds for both terms, the total objective is to minimize the upper bound. After dropping the constant term $\sigma^2 n$, which does not affect the solution, the objective becomes minimizing $\sum_{i\in[1:k]} \delta_i(\sigma^2 \sum_{j\in[1:n]} (\alpha_i^j)^2 + C(\sum_{j\in[1:n]} \alpha_i^j)^2)$ subject to the unbiasedness constraint $\sum_{i\in[1:k]} \alpha_i^j = 1$.

This reformulated optimization problem, adapted to the mini-batch SGD setting, can be explicitly solved using the KKT conditions. This yields a dense, closed-form solution:

$$(\alpha_i^j)^* = \frac{\delta_i^{-1}}{\sum_{l\in[1:k]} \delta_l^{-1}}, \forall i \in [1:k], j \in [1:n]. \tag{41}$$

While this solution is theoretically optimal, it requires full data replication across all worker nodes ($d = k$), meaning that each node must store the entire dataset. This assumption is often impractical in distributed systems due to excessive storage and computation costs.

On the other hand, our proposed schemes (Schemes I and II), which are formulated and constructed based on the GD algorithm, are specifically designed to maintain a low data replication factor, ensuring $d < 2$ in both schemes. When our proposed schemes, which are optimized for the full-batch GD setting, are directly applied to mini-batch SGD, the resulting gradient estimator does not achieve the theoretical minimum residual error with respect to the true gradient. Nonetheless, our approach maintains the important advantage of significantly reduced data replication under the mini-batch SGD setting.

This observation highlights a trade-off when applying mini-batch SGD in distributed learning. Achieving the theoretically minimum residual error requires each worker node to handle a substantially increased computation load, which can accelerate convergence. In contrast, by utilizing our optimally constructed GD-based gradient codes and applying mini-batch sampling over the data partitions assigned to each worker, one may not reach the theoretical minimum of the residual error, but can substantially reduce the computation load per worker node. Note that since the mini-batch SGD process introduces an inherent variance from sampling, our gradient coding scheme, which is optimized for the GD algorithm, does not perfectly mitigate this particular sampling variance.

In particular, implementing mini-batch SGD using our proposed schemes results in a residual error bounded by $n^2 C \cdot \frac{1}{\sum_{i=1}^{k} \delta_i^{-1}} + n\sigma^2\left(1 + \frac{n}{\sum_{i=1}^{k} \delta_i^{-1}}\right)$. This introduces an additional term, $n\sigma^2(1 + \frac{n}{\sum_{i=1}^{k} \delta_i^{-1}})$, due to batch sampling, compared to the result in Lemma 2 of the paper; however, within our convergence analysis (Theorems 3, 2, and 4), the scaling term $n^2 C \cdot (1 + \frac{1}{\sum_{i=1}^{k} \delta_i^{-1}})$ is only slightly modified to $n^2 C \cdot (1 + \frac{1}{\sum_{i=1}^{k} \delta_i^{-1}}) + n\sigma^2(1 + \frac{n}{\sum_{i=1}^{k} \delta_i^{-1}})$ and thus the convergence rate remains essentially unchanged.

### C.4.2 Extension to Adaptive Gradient Method

Our proposed method improves convergence speed by ensuring unbiasedness in the gradient estimator while reducing variance. This has been effective for gradient-descent-like optimizers, which use only the first moment estimator for updates. However, in adaptive gradient methods, which also utilize the squared gradients for second-moment estimation, maintaining unbiasedness while reducing variance faces inherent limitations in decreasing the variance term of the second moment. The reason is that preserving unbiasedness inevitably causes $\mathbb{E}[\|g - \hat{g}\|_2^2]$ to accumulate as variance. This, in turn, systematically inflates the denominator in Adam's update rule, leading to excessive

step-size shrinkage and possible performance degradation. In summary, this is a manifestation of the bias–variance tradeoff: forcing bias to zero can hinder squared gradient estimation, which inherently contains both bias and variance components.

To address this, we additionally propose a two-track decoding approach. This scheme preserves the first-moment estimation of the original method while, for the second moment, allowing a slight bias to reduced the variance, thereby leading to a more accurate second-moment estimate overall. The implementation introduces negligible overhead, as the encoding remains unchanged and the master node simply uses two decoding vectors during decoding.

Since the encoder is fixed, we focus on designing the decoder. Designing a decoder that reduces both bias and variance leads to solving the following problem, where $v$ is used instead of $w$ for notational clarity:

$$\min_v \Lambda[\sum_j(1 - \sum_i(1 - p_i)v_ia_{i,j})]^2 + [\sum_i p_i(1 - p_i)v_i^2(\sum_j a_{i,j})^2]. \tag{42}$$

The first term corresponds to bias reduction, the second to variance reduction, and $\Lambda$ determines the relative emphasis on reducing bias. The solution can be derived by:

$$v_i^* = \frac{n\Lambda}{p_i(\sum_{j=1}^n a_{i,j})(1 + \Lambda \sum_{m=1}^k \delta_m^{-1})}. \tag{43}$$

Using the proposed Schemes I or II to generate the encoding matrix $A$, the master node applies the original $w$ for the first-moment estimation and the above $v$ for the second-moment estimation, decoding each separately to obtain the gradient estimators. These are then used to update the model following the update rule of optimizers such as Adam. In Appendix E, Figure 7 demonstrates that because conventional gradient coding techniques are designed for gradient-descent-like methods, which primarily rely on the first moment, they face inherent challenges in adaptive gradient methods that also require a stable second moment. The results also highlight the effectiveness of the two-track decoding technique for these adaptive methods, indicating a need for further research and analysis into gradient coding schemes that are compatible with various optimizers.

### C.4.3  Extension to Non-Smooth Loss Function

For the non-smooth case, while not explicitly detailed in the paper, our method's convergence is guaranteed under the well-established analysis in [22]. Specifically, the convergence analysis presented in [22] holds as long as two key assumptions are satisfied—both of which are met by our method:

- **Unbiased estimator**: Our method is designed to provide an unbiased estimator of the gradient ($g^{(t)}$), as enforced by the condition in Equation (12).
- **Bounded variance**: Our optimally structured coding scheme ensures a formal upper bound on the estimator's variance, as proven in Lemma 2.

Since our algorithm meets these conditions, the convergence guarantees established by [22] for non-smooth settings can directly apply to our method. This confirms the theoretical robustness of our proposed method for a broad range of practical applications.

### C.5  Limitations

One important limitation of our approach is the substantial communication overhead incurred during the initial distribution of large datasets: because each data partition $\mathcal{D}_i$ must be replicated to $d_i$ worker nodes before training begins, network load grows with both dataset size and replication factor. Additionally, our design relies on knowing each worker node's straggler probability $p_i$ a priori, yet obtaining reliable estimates is challenging because existing straggler models cannot fully capture the diverse, interacting factors—such as network congestion, CPU/GPU heterogeneity, energy throttling, or transient OS interrupts—that actually cause worker nodes to slow down or drop out. Exploring and validating more practical, multifactor straggler models therefore remains an important avenue for future work.

Nonetheless, to overcome this limitation, a practical estimation approach can be applied in a real-world distributed learning environment:

- The most practical approach is for the master node to estimate each worker's straggler probability by simply counting how often a worker exceeds the deadline, based on historical logs, and updating this estimate periodically. For example, the master can track how many times each worker was late out of the most recent tasks, and use this frequency as the current estimate of the straggler probability. This method is commonly known as empirical maximum likelihood estimation (MLE).

- Another practical approach is to estimate each worker node's straggling probability $p_i$ using a well-modeled estimator, as proposed in [18, 19]. Specifically, task completion times can be modeled using parametric probability distributions, such as the shifted-exponential distribution employed in [18]. By fitting this distribution to observed runtime data, we obtain reliable estimates of $p_i, \forall i$, representing the probability that worker $i$ fails to meet the deadline. These estimated values can then be directly integrated into our coding scheme as fixed parameters. Experimental results in [18], conducted on a real EC2 cluster, demonstrate the effectiveness of this method in enhancing the speed and robustness of distributed learning systems. In our own experiments, we adopted this same estimation approach for $p_i$ and based our evaluations on the resulting values.

## D Implementation Details

The experiments are conducted on one NVIDIA GeForce RTX 3060 GPU (12 GB), six NVIDIA GeForce GTX 1080 GPUs (8 GB each), and twelve NVIDIA Tesla P100 GPUs (16 GB each) (provided through Kaggle Cloud). Due to the memory limit, we use the gradient accumulation technique, which divides each training batch into smaller sub-batches processed sequentially; gradients are accumulated over these steps before performing a single parameter update, effectively simulating a larger batch size without exceeding GPU memory constraints.

We compare our design with benchmarks as follows:

- **GD**: GD updates parameters using the true gradient $g^{(t)}$, which makes it immune to straggler effects and represents an ideal centralized learning scenario.

- **IS-SGD** (Ignore-Stragglers SGD): IS-SGD assigns disjoint data partitions to each node (i.e., $d = 1$) to avoid redundancy, yet it remains subject to straggler effects, which it does not mitigate but essentially ignores.

- **BGC [8]** (Bernoulli Gradient Coding): In BGC, each encoding coefficient is generated according to a Bernoulli distribution, i.e., $a_{i,j} \sim \text{Bernoulli}(d/k)$ for all $i, j$, while all decoding coefficients are fixed to $w_i = 1$.

- **EHD [9]** (ERASUREHEAD): EHD constructs encoding coefficients using the FR code [5], and sets the decoding coefficients uniformly to $w_i = 1$ for all nodes.

- **OD [11]** (Optimal Decoding): OD determines the encoding coefficients $a_{i,j} \in \{0, 1\}$ using a random graph, and dynamically computes optimal decoding coefficients $w_i$ for the straggler effect reduction in each iteration.

- **SGC [12]** (Stochastic Gradient Coding): SGC employs the pair-wise data distribution strategy to build an unbiased gradient estimator, with redundancy determined as in [12]. To accommodate heterogeneous straggler scenarios, we modify the approach by setting $a_{i,j} = \frac{1}{d_j(1-p_i)}$ for all $i, j$, while keeping $w_i = 1$ for all nodes.

The methods BGC, EHD, OD, and SGC incorporate the data redundancy of approximately $d \approx 2$.

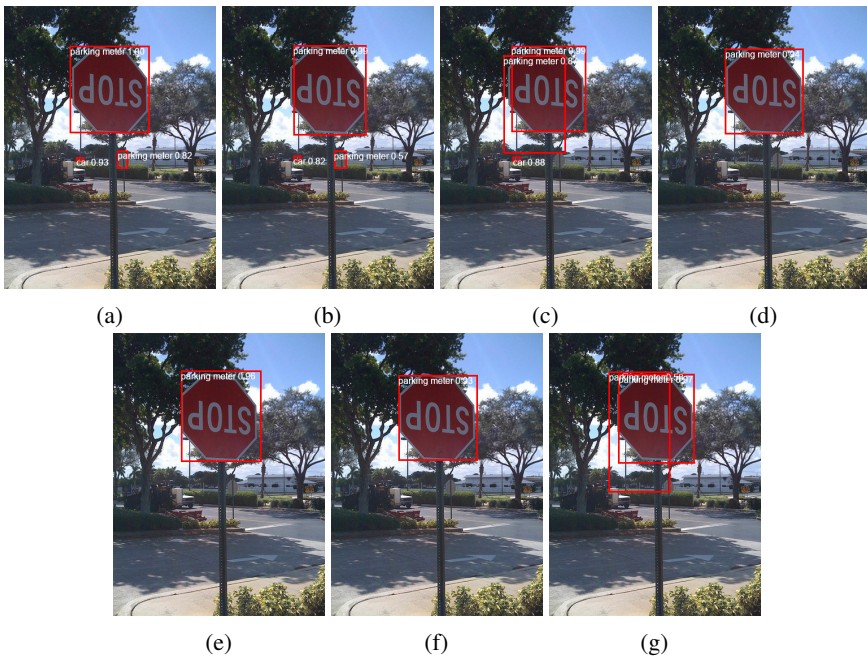

(a)  (b)  (c)  (d)

(e)  (f)  (g)

Figure 5: Detected objects of sampled image: (a) GD (b) Proposed (c) SGC (d) EHD (e) BGC (f) OD (g) IS-SGD.

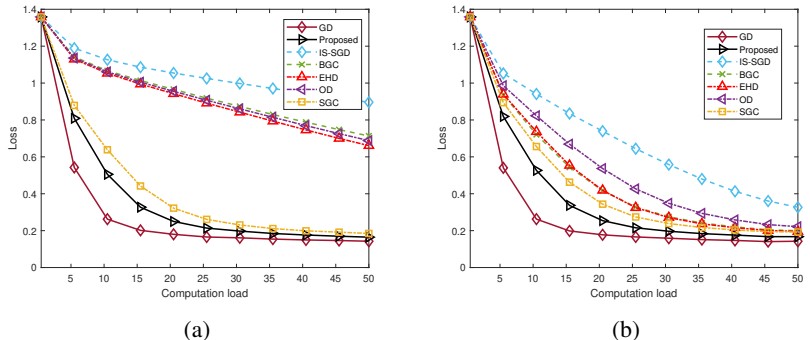

(a)  (b)

Figure 6: Convergence graph with respect to the computation load $d$ with RetinaNet: (a) $\tau_{th} = 1.1$ (b) $\tau_{th} = 1.5$.

# E  Additional Experiment Results

Figure 5 displays the detection results on a representative image from the COCO validation set. The straggler-free GD identifies three objects—two parking meters and one car—and the proposed design detects exactly the same three objects, with no extras. In contrast, SGC erroneously splits one parking meter into two overlapping detections and therefore returns only one car and one (duplicated) parking meter. EHD, BGC, and OD each find just a single parking meter, missing the car and another parking meter, while IS-SGD mistakes one parking meter for two separate instances. These results demonstrate that our method delivers GD-level detection quality while simultaneously neutralising straggler effects.

Since the achieved performance depends on the specific model or dataset employed, we have additionally conducted supplementary experiments using the state-of-the-art RetinaNet model to benchmark performance. RetinaNet, with approximately 34 million parameters, is roughly 6.3 times larger than the model (5.4 million parameters) employed in our experiments (Section 4). Figures 6(a) and 6(b) illustrates the convergence behavior with respect to the computation load $d$ for $k = 10$

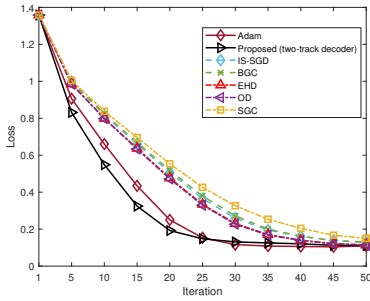

Figure 7: Convergence graph with respect to the training iteration $T$ with Adam optimizer ($\tau_{th} = 1.1$ and $\gamma_t = 0.001$).

and $\tau_{th} = 1.1$ and $1.5$, respectively. Based on these results, it can be seen that the proposed method consistently maintains the performance trends identified in Section 4, regardless of the model size.

Moreover, in Figure 7, we evaluate the convergence behavior with respect to the training iteration $T$ to assess the performance of the two-track decoder with $\Lambda = 1$ detailed in Appendix C.4.2. We observe that gradient coding schemes for the first moment exhibit a slower convergence speed compared to centralized Adam. However, the performance gap between the baselines is notably smaller with the Adam optimizer than with GD. This reduced difference is primarily due to Adam's intrinsic noise suppression effect. In contrast, the proposed two-track decoding mitigates the second-moment estimation error inherent in conventional gradient coding. While this approach yields significant performance improvements and shows extendability, a more comprehensive analysis of these results is required in future research.

# F Proofs

## F.1 Proof of Lemma 1

From the equations (11)-(12), we have

$$\mathbb{E}_t[\|g^{(t)} - \hat{g}^{(t)}\|_2^2] = \mathbb{E}_t\left[\left\|\sum_{j=1}^{n} g_j^{(t)}\left(1 - \sum_{i=1}^{k} \mathbb{I}_i \cdot w_i a_{i,j}\right)\right\|_2^2\right] \tag{44}$$

$$= \sum_{j_1=1}^{n} \sum_{j_2=1}^{n} \langle g_{j_1}^{(t)}, g_{j_2}^{(t)} \rangle \cdot \mathbb{E}_t\left[\left(1 - \sum_{i=1}^{k} \mathbb{I}_i \cdot w_i a_{i,j_1}\right)\left(1 - \sum_{i=1}^{k} \mathbb{I}_i \cdot w_i a_{i,j_2}\right)\right] \tag{45}$$

$$\overset{(a)}{\leq} C \sum_{j_1=1}^{n} \sum_{j_2=1}^{n} \mathbb{E}_t\left[1 - \sum_{i=1}^{k} \mathbb{I}_i \cdot w_i a_{i,j_1} - \sum_{i=1}^{k} \mathbb{I}_i \cdot w_i a_{i,j_2} + \left(\sum_{i=1}^{k} \mathbb{I}_i \cdot w_i a_{i,j_1}\right)\left(\sum_{i=1}^{k} \mathbb{I}_i \cdot w_i a_{i,j_2}\right)\right] \tag{46}$$

$$\overset{(b)}{=} C \sum_{j_1=1}^{n} \sum_{j_2=1}^{n} \mathbb{E}_t\left[\left(\sum_{i=1}^{k} \mathbb{I}_i \cdot w_i a_{i,j_1}\right)\left(\sum_{i=1}^{k} \mathbb{I}_i \cdot w_i a_{i,j_2}\right) - 1\right], \tag{47}$$

where (a) and (b) come from the boundedness assumption of the gradient and the unbiasedness of the gradient estimator, respectively. Furthermore,

$$C \sum_{j_1=1}^{n} \sum_{j_2=1}^{n} \mathbb{E}_t\left[-1 + \left(\sum_{i=1}^{k} \mathbb{I}_i \cdot w_i a_{i,j_1}\right)\left(\sum_{i=1}^{k} \mathbb{I}_i \cdot w_i a_{i,j_2}\right)\right] \tag{48}$$

$$\overset{(c)}{=} C \sum_{j_1=1}^{n} \sum_{j_2=1}^{n} \left[-1 + \mathbb{E}_t\left[\sum_{i_1=1}^{k} \sum_{i_2=1, i_1 \neq i_2}^{k} \mathbb{I}_{i_1} \mathbb{I}_{i_2} w_{i_1} w_{i_2} a_{i_1,j_1} a_{i_2,j_2} + \sum_{i=1}^{k} \mathbb{I}_i \cdot w_i^2 a_{i,j_1} a_{i,j_2}\right]\right] \tag{49}$$

$$\overset{(d)}{=} C \sum_{j_1=1}^{n} \sum_{j_2=1}^{n} \left[-\sum_{i=1}^{k} (1 - p_i)^2 \cdot w_i^2 a_{i,j_1} a_{i,j_2} + \sum_{i=1}^{k} (1 - p_i) \cdot w_i^2 a_{i,j_1} a_{i,j_2}\right], \tag{50}$$

where (c) is because $\mathbb{E}_t[\mathbb{I}_i \cdot \mathbb{I}_i] = \mathbb{E}_t[\mathbb{I}_i]$, and (d) is due to

$$\mathbb{E}_t\left[\sum_{i_1=1}^{k} \sum_{i_2=1, i_1 \neq i_2}^{k} \mathbb{I}_{i_1} \mathbb{I}_{i_2} \cdot w_{i_1} w_{i_2} a_{i_1,j_1} a_{i_2,j_2}\right] = \sum_{i_1=1}^{k} \sum_{i_2=1, i_1 \neq i_2}^{k} (1 - p_{i_1})(1 - p_{i_2}) w_{i_1} w_{i_2} a_{i_1,j_1} a_{i_2,j_2} \tag{51}$$

$$= \left(\sum_{i_1=1}^{k} (1 - p_{i_1}) \cdot w_{i_1} a_{i_1,j_1}\right)\left(\sum_{i_2=1}^{k} (1 - p_{i_2}) \cdot w_{i_2} a_{i_2,j_2}\right) - \sum_{i=1}^{k} (1 - p_i)^2 \cdot w_i^2 a_{i,j_1} a_{i,j_2} \tag{52}$$

$$= 1 - \sum_{i=1}^{k} (1 - p_i)^2 \cdot w_i^2 a_{i,j_1} a_{i,j_2}. \tag{53}$$

Putting together, we have

$$\mathbb{E}_t[\|g^{(t)} - \hat{g}^{(t)}\|_2^2] \leq C\left[\sum_{i=1}^{k} p_i(1 - p_i) \cdot w_i^2 \left(\sum_{j=1}^{n} a_{i,j}\right)^2\right]. \tag{54}$$

## F.2 Proof of theorem 1

The Lagrangian function of (**P3**) is

$$\mathcal{L}(\boldsymbol{\alpha}) = \sum_{i=1}^{k} \delta_i \left(\sum_{j=1}^{n} \alpha_i^j\right)^2 + \sum_{j=1}^{n} \zeta_j \left(\sum_{i=1}^{k} \alpha_i^j - 1\right), \tag{55}$$

where $\zeta_j$ is a Lagrangian multiplier.

By Karush-Kuhn-Tucker (KKT) conditions, we have

$$\begin{cases} \frac{\partial \mathcal{L}}{\partial \alpha_i^j} = 2\delta_i \left( \sum_{j=1}^n \alpha_i^j \right) + \zeta_j = 0, \forall i, j, \\ \sum_{i=1}^k \alpha_i^j = 1, \forall j. \end{cases} \tag{56}$$

From the KKT condition on stationarity, we have

$$\delta_1 \left( \sum_{j=1}^n \alpha_1^j \right) = \delta_2 \left( \sum_{j=1}^n \alpha_2^j \right) = \cdots = \delta_n \left( \sum_{j=1}^n \alpha_n^j \right). \tag{57}$$

Then, let $X = \delta_i \sum_{j=1}^n \alpha_i^j, \forall i$ and using the primal feasibility, i.e., $\sum_{i=1}^k \alpha_i^j = 1, \forall j$, we have

$$\sum_{i=1}^k \left( X \cdot \delta_i^{-1} \right) = \sum_{i=1}^k \left( \sum_{j=1}^n \alpha_i^j \right) = n, \tag{58}$$

and thus,

$$X = \frac{n}{\sum_{i=1}^k \delta_i^{-1}}. \tag{59}$$

Accordingly, the optimal gradient codes, which minimize the gradient estimation error under the unbiasedness constraint, can be obtained when matrix $\boldsymbol{\alpha}$ satisfies the following conditions:

$$\sum_{j=1}^n (\alpha_i^j)^* = Y_i, \forall i, \text{ and } \sum_{i=1}^k (\alpha_i^j)^* = 1, \forall j, \tag{60}$$

where $(\alpha_i^j)^*$ is the optimal $\alpha_i^j$, and $Y_i = \delta_i^{-1} \cdot \frac{n}{\sum_{j=1}^k \delta_j^{-1}}$.

### F.3   Proof of lemma 2

From Lemma 1 and Theorem 1, we can easily derive the bounded residual error of gradient estimator for any optimally structured gradient codes in the following:

$$\mathbb{E}_t[\|g^{(t)} - \hat{g}^{(t)}\|_2^2] \leq C \left[ \sum_{i=1}^k \delta_i \cdot \left( \sum_{j=1}^n (\alpha_i^j)^* \right)^2 \right] \tag{61}$$

$$\leq n^2 C \cdot \frac{1}{\sum_{i=1}^k \delta_i^{-1}}, \tag{62}$$

where $\alpha_i^j = \tilde{w}_i a_{i,j}$, $(\alpha_i^j)^*$ represents the optimal $\alpha_i^j$, and $\delta_i^{-1} = \frac{1 - p_i}{p_i}$. Furthermore, the squared norm of the gradient estimator for any optimally structured gradient codes is bounded by

$$\mathbb{E}_t[\|\hat{g}^{(t)}\|_2^2] = \mathbb{E}_t[\|g^{(t)} - \hat{g}^{(t)}\|_2^2] + \|g^{(t)}\|_2^2 \tag{63}$$

$$\leq C \left[ n^2 + \sum_{i=1}^k \delta_i \cdot \left( \sum_{j=1}^n (\alpha_i^j)^* \right)^2 \right] \tag{64}$$

$$= n^2 C \cdot \left( 1 + \frac{1}{\sum_{i=1}^k \delta_i^{-1}} \right). \tag{65}$$

### F.4   Proof of theorem 2

Our proof builds upon the result from [20], which demonstrates that any algorithm utilizing an unbiased estimator of the true gradient achieves a convergence rate of $O(1/T)$:

**Lemma 3.** *(Lemma 1 in [20])  Suppose the loss function is $\lambda$-strongly convex and the gradient estimator is unbiased. Furthermore, assume $\mathbb{E}_t[\|\hat{g}^{(t)}\|_2^2] \leq G$. Then, by setting $\gamma_t = 1/(\lambda t)$, the following holds for any $T$ that*

$$\mathbb{E}[\|\boldsymbol{\beta}_T - \boldsymbol{\beta}^*\|_2^2] \leq \frac{4G}{\lambda T}. \tag{66}$$

Building on the result from Lemma 2, we can conclude that

$$\mathbb{E}_t[\|\hat{g}^{(t)}\|_2^2] \leq n^2 C \cdot \left(1 + \frac{1}{\sum_{i=1}^k \delta_i^{-1}}\right). \tag{67}$$

Thus, by replacing $G$ with the right-hand side of the above inequality, Lemma 3 yields

$$\mathbb{E}[\|\boldsymbol{\beta}_T - \boldsymbol{\beta}^*\|_2^2] \leq \frac{4n^2 C}{\lambda^2 T}\left(1 + \frac{1}{\sum_{i=1}^k \delta_i^{-1}}\right), \tag{68}$$

where $\delta_i^{-1} = \frac{1-p_i}{p_i}$.

### F.5 Proof of theorem 3

From the property of $\mu$-smoothness,

$$L(\boldsymbol{\beta}_{t+1}) = L(\boldsymbol{\beta}_t - \gamma_t \cdot \hat{g}^{(t)}) \tag{69}$$

$$\leq L(\boldsymbol{\beta}_t) - \langle g^{(t)}, \gamma_t \cdot \hat{g}^{(t)}\rangle + \frac{\mu\gamma_t^2}{2}\|\hat{g}^{(t)}\|_2^2. \tag{70}$$

By taking the expectation $\mathbb{E}_t[\cdot]$ conditioned on the previous iteration on both hand sides, we have

$$\mathbb{E}_t[L(\boldsymbol{\beta}_{t+1})] \leq L(\boldsymbol{\beta}_t) - \langle g^{(t)}, \gamma_t \cdot \mathbb{E}_t[\hat{g}^{(t)}]\rangle + \frac{\mu\gamma_t^2}{2}\mathbb{E}_t[\|\hat{g}^{(t)}\|_2^2] \tag{71}$$

$$\overset{(a)}{\leq} L(\boldsymbol{\beta}_t) - \gamma_t \cdot \|g^{(t)}\|_2^2 + \frac{\mu\gamma_t^2 n^2 C}{2}\left(1 + \frac{1}{\sum_{i=1}^k \delta_i^{-1}}\right), \tag{72}$$

where (a) comes from the unbiasedness of gradient estimator and Lemma 2. Taking full expectation $\mathbb{E}[\mathbb{E}_t[\cdot]]$ on both sides and rearranging, we obtain

$$\gamma_t \cdot \mathbb{E}[\|g^{(t)}\|_2^2] \leq \mathbb{E}[L(\boldsymbol{\beta}_t)] - \mathbb{E}[L(\boldsymbol{\beta}_{t+1})] + \frac{\mu\gamma_t^2 n^2 C}{2} \cdot \left(1 + \frac{1}{\sum_{i=1}^k \delta_i^{-1}}\right). \tag{73}$$

Based on this inequality, we have

$$\sum_{t=0}^T \gamma_t \cdot \mathbb{E}[\|g^{(t)}\|_2^2] \leq L(\boldsymbol{\beta}_0) - \mathbb{E}[L(\boldsymbol{\beta}_{T+1})] + \frac{\mu n^2 C}{2} \cdot \left(1 + \frac{1}{\sum_{i=1}^k \delta_i^{-1}}\right)\sum_{t=0}^T \gamma_t^2 \tag{74}$$

$$\overset{(b)}{\leq} L(\boldsymbol{\beta}_0) - L(\boldsymbol{\beta}^*) + \frac{\mu n^2 C}{2} \cdot \left(1 + \frac{1}{\sum_{i=1}^k \delta_i^{-1}}\right)\sum_{t=0}^T \gamma_t^2, \tag{75}$$

where (b) is due to $\mathbb{E}[L(\boldsymbol{\beta}_{T+1})] \geq L(\boldsymbol{\beta}^*)$.

If the learning rate is fixed, i.e., $\gamma_t = \gamma = 1/(T+1)^{1/2}$, we have

$$\frac{1}{T+1}\sum_{t=0}^T \mathbb{E}[\|g^{(t)}\|_2^2] \leq \frac{L(\boldsymbol{\beta}_0) - L(\boldsymbol{\beta}^*)}{(T+1)^{1/2}} + \frac{1}{(T+1)^{1/2}}\frac{\mu n^2 C}{2} \cdot \left(1 + \frac{1}{\sum_{i=1}^k \delta_i^{-1}}\right), \tag{76}$$

where the following limit holds:

$$\lim_{T\to\infty} \frac{L(\boldsymbol{\beta}_0) - L(\boldsymbol{\beta}^*) + \frac{\mu n^2 C}{2} \cdot \left(1 + \frac{1}{\sum_{i=1}^k \delta_i^{-1}}\right)}{(T+1)^{1/2}} = 0. \tag{77}$$

Moreover, if the learning rate is decaying, i.e., $\gamma_t = 1/(t+1)^{1/2}$, we have

$$\sum_{t=0}^T \frac{1}{(t+1)^{1/2}} \cdot \mathbb{E}[\|g^{(t)}\|_2^2] \geq \frac{1}{(T+1)^{1/2}}\sum_{t=0}^T \mathbb{E}[\|g^{(t)}\|_2^2]. \tag{78}$$

Thus, using this relations,

$$\frac{1}{T+1}\sum_{t=0}^{T}\mathbb{E}[\|g^{(t)}\|_2^2] \leq \frac{L(\boldsymbol{\beta}_0) - L(\boldsymbol{\beta}^*)}{(T+1)^{1/2}} + \frac{1}{(T+1)^{1/2}}\frac{\mu n^2 C}{2}\cdot\left(1 + \frac{1}{\sum_{i=1}^{k}\delta_i^{-1}}\right)\sum_{t=0}^{T}\frac{1}{t+1} \tag{79}$$

$$\stackrel{(c)}{\leq} \frac{L(\boldsymbol{\beta}_0) - L(\boldsymbol{\beta}^*)}{(T+1)^{1/2}} + \frac{\mu n^2 C(1 + \log(T+1)^{1/2})\left(1 + \frac{1}{\sum_{i=1}^{k}\delta_i^{-1}}\right)}{(T+1)^{1/2}}, \tag{80}$$

where (c) is due to the fact $\sum_{t=0}^{T}\frac{1}{t+1} \leq 2 + \log(T+1)$. Since $\lim_{x\to\infty}\frac{\log x}{x} = 0$, the following limit holds:

$$\lim_{T\to\infty}\frac{L(\boldsymbol{\beta}_0) - L(\boldsymbol{\beta}^*)}{(T+1)^{1/2}} + \frac{\mu n^2 C(1 + \log(T+1)^{1/2})\left(1 + \frac{1}{\sum_{i=1}^{k}\delta_i^{-1}}\right)}{(T+1)^{1/2}} = 0. \tag{81}$$

### F.6   Proof of theorem 4

Since $\boldsymbol{\beta}_{t+1} = \boldsymbol{\beta}_t - \gamma_t\cdot\hat{g}^{(t)}$,

$$\|\boldsymbol{\beta}_{t+1} - \boldsymbol{\beta}^*\|_2^2 = \|\boldsymbol{\beta}_t - \boldsymbol{\beta}^* - \gamma_t\cdot\hat{g}^{(t)}\|_2^2 \tag{82}$$

Then, by taking expectation of both side conditioned on $\boldsymbol{\beta}_t$,

$$\mathbb{E}_t[\|\boldsymbol{\beta}_{t+1} - \boldsymbol{\beta}^*\|_2^2] = \mathbb{E}_t[\|\boldsymbol{\beta}_t - \boldsymbol{\beta}^* - \gamma_t\cdot\hat{g}^{(t)}\|_2^2] \tag{83}$$

$$= \|\boldsymbol{\beta}_t - \boldsymbol{\beta}^*\|_2^2 - 2\gamma_t\cdot\mathbb{E}_t[(\boldsymbol{\beta}_t - \boldsymbol{\beta}^*)^\top\hat{g}^{(t)}] + \gamma_t^2\cdot\mathbb{E}_t[\|\hat{g}^{(t)}\|_2^2] \tag{84}$$

$$\stackrel{(a)}{\leq} \|\boldsymbol{\beta}_t - \boldsymbol{\beta}^*\|_2^2 + 2\gamma_t\cdot\left(L(\boldsymbol{\beta}^*) - L(\boldsymbol{\beta}_t) - \frac{\lambda}{2}\|\boldsymbol{\beta}_t - \boldsymbol{\beta}^*\|_2^2\right) + \gamma_t^2\cdot n^2 C\cdot\left(1 + \frac{1}{\sum_{i=1}^{k}\delta_i^{-1}}\right) \tag{85}$$

$$= (1 - \gamma_t\lambda)\cdot\|\boldsymbol{\beta}_t - \boldsymbol{\beta}^*\|_2^2 + 2\gamma_t\cdot(L(\boldsymbol{\beta}^*) - L(\boldsymbol{\beta}_t)) + \gamma_t^2\cdot n^2 C\cdot\left(1 + \frac{1}{\sum_{i=1}^{k}\delta_i^{-1}}\right) \tag{86}$$

$$\stackrel{(b)}{\leq} (1 - \gamma_t\lambda)\cdot\|\boldsymbol{\beta}_t - \boldsymbol{\beta}^*\|_2^2 - \frac{\gamma_t}{\mu}\|g^{(t)}\|_2^2 + \gamma_t^2\cdot n^2 C\cdot\left(1 + \frac{1}{\sum_{i=1}^{k}\delta_i^{-1}}\right) \tag{87}$$

$$\leq (1 - \gamma_t\lambda)\cdot\|\boldsymbol{\beta}_t - \boldsymbol{\beta}^*\|_2^2 + \gamma_t^2\cdot n^2 C\cdot\left(1 + \frac{1}{\sum_{i=1}^{k}\delta_i^{-1}}\right), \tag{88}$$

where (a) holds from the $\lambda$-strongly convexity and Lemma 2; (b) is due to $\mu$-smoothness of the loss function $L$. Specifically,

$$L\left(\boldsymbol{\beta}_t - \frac{1}{\mu}\nabla L(\boldsymbol{\beta}_t)\right) \leq L(\boldsymbol{\beta}_t) + \left\langle\nabla L(\boldsymbol{\beta}_t), -\frac{1}{\mu}\nabla L(\boldsymbol{\beta}_t)\right\rangle + \frac{\mu}{2}\left\|-\frac{1}{\mu}\nabla L(\boldsymbol{\beta}_t)\right\|_2^2 \tag{89}$$

$$= L(\boldsymbol{\beta}_t) - \frac{1}{2\mu}\left\|\nabla L(\boldsymbol{\beta}_t)\right\|_2^2. \tag{90}$$

Using the relationship $L(\boldsymbol{\beta}^*) \leq L\left(\boldsymbol{\beta}_t - \frac{1}{\mu}\nabla L(\boldsymbol{\beta}_t)\right)$,

$$L(\boldsymbol{\beta}^*) - L(\boldsymbol{\beta}_t) \leq -\frac{1}{2\mu}\left\|\nabla L(\boldsymbol{\beta}_t)\right\|_2^2. \tag{91}$$

Taking full expectation of both side in (87), we have

$$\mathbb{E}[\|\boldsymbol{\beta}_{t+1} - \boldsymbol{\beta}^*\|_2^2] \leq (1 - \gamma_t\lambda)\cdot\mathbb{E}[\|\boldsymbol{\beta}_t - \boldsymbol{\beta}^*\|_2^2] + \gamma_t^2\cdot n^2 C\cdot\left(1 + \frac{1}{\sum_{i=1}^{k}\delta_i^{-1}}\right). \tag{92}$$

Then, using this inequality recursively, the following inequality holds.

$$\mathbb{E}[\|\boldsymbol{\beta}_{t+1}-\boldsymbol{\beta}^*\|_2^2] \le \|\boldsymbol{\beta}_0-\boldsymbol{\beta}^*\|_2^2 \cdot \prod_{p=0}^{t}(1-\gamma_p\lambda)+n^2C\cdot\left(1+\frac{1}{\sum_{i=1}^{k}\delta_i^{-1}}\right)\cdot\left\{\gamma_t^2+\sum_{p=0}^{t-1}\gamma_p^2\prod_{q=p+1}^{t}(1-\gamma_q\lambda)\right\}.$$

(93)

If $\gamma_t = \gamma < 1/\lambda$, $\forall t$ and $t \to \infty$,

$$\lim_{t\to\infty}\mathbb{E}[\|\boldsymbol{\beta}_{t+1}-\boldsymbol{\beta}^*\|_2^2] \le \lim_{t\to\infty}\|\boldsymbol{\beta}_0-\boldsymbol{\beta}^*\|_2^2 \cdot (1-\gamma\lambda)^{t+1}+\frac{\gamma n^2 C}{\lambda}\left(1+\frac{1}{\sum_{i=1}^{k}\delta_i^{-1}}\right)(1-(1-\gamma\lambda)^{t+1})$$

(94)

$$< \frac{n^2C}{\lambda^2}\cdot\left(1+\frac{1}{\sum_{i=1}^{k}\delta_i^{-1}}\right).$$

(95)

Furthermore, if $\gamma_t = 1/(\lambda t)$ and $t \to \infty$,

$$\lim_{t\to\infty}\mathbb{E}[\|\boldsymbol{\beta}_{t+1}-\boldsymbol{\beta}^*\|_2^2] \le \lim_{t\to\infty}\mathbb{E}[\|\boldsymbol{\beta}_1-\boldsymbol{\beta}^*\|_2^2] \cdot \prod_{p=1}^{t}\left(1-\frac{1}{p}\right)+n^2C\cdot\left(1+\frac{1}{\sum_{i=1}^{k}\delta_i^{-1}}\right)\times$$

$$\left\{\sum_{p=1}^{t-1}\frac{1}{\lambda^2 p^2}\prod_{q=p+1}^{t}\left(1-\frac{1}{q}\right)+\frac{1}{\lambda^2 t^2}\right\}$$

(96)

$$= \lim_{t\to\infty}O\left(\frac{1}{t}\right)$$

(97)

$$= 0.$$

(98)

