# OpenReview forum: "Approximate Gradient Coding for Distributed Learning with Heterogeneous Stragglers"
_NeurIPS.cc/2025/Conference — NeurIPS 2025 poster_

### Official Review · Reviewer_dCZy · 2025-06-29

**Clarity:** 4
**Significance:** 3
**Originality:** 3
**Rating:** 4
**Confidence:** 3

**Summary:**

This paper propose a gradient coding scheme for distributed machine learning, which minimizes residual error and maintaining an unbiased gradient estimator. Theoretical analysis and simulations are provided.

**Questions:**

**Questions**:

1. How does the paper get the conclusion that (P2) is non-convex and (P3) is convex quickly? Can you provide more details. Are there some rules to help you get these conclusions quickly. I am curious about this.

2. Lines 219-220:

   > This design reduces the overall computation load on individual worker nodes.

   Can you kindly explain this in detail?

**Suggestions**:

1. The font size of the appendix seems smaller than the font size of the main body. It may be better to use the same font size for both.
2. Equation (61): For me, Equation (61) is unnecessary. From Equation (60) to Equation (62) is straightforward by Equations (58) and (59).

I cannot guarantee a high score (for example, 5 and 6), as I am not sure whether this simple way is appreciated by the experts in gradient coding.

**Ethical Concerns:**

["NO or VERY MINOR ethics concerns only"]

**Final Justification:**

The rebuttal addressed my concerns.

Given that this paper is solid, simple and clear, I maintain my positive rating.

I hope that the authors will revise the paper as promised to improve the readability further.

As said in my initial review, I am unfamiliar with gradient coding. This is the only reason that I did not give a higher rating.

**Limitations:**

yes

**Paper Formatting Concerns:**

No.

**Quality:**

3

**Strengths And Weaknesses:**

**Strengths**:

1. This paper proposed one method that minimizes the residual error and ensures the gradient estimator's unbiasedness.
2. This paper is very well-written, well-organized.
3. This paper is self-contained and very clear. This is the first paper I read about gradient coding, however, it is very easy for me to follow the main idea of this paper.

**Weaknesses**:

After checking all the proofs (except the convergence analysis, i.e., Theorems 2 and 3) carefully, I have not found any serious weaknesses.

---

> ### Author Rebuttal · Authors · 2025-07-30
>
> Thank you for the detailed review. We appreciate the time and energy you put on this work. We reply to your questions one by one.
>
> ***[Question-1]***
>
> The conclusions are drawn from standard principles of optimization theory.
> - For (P2): The of non-convexity is the presence of bilinear terms involving products of optimization variables, specifically $w_i a_{i,j}$. Bilinear terms generally lead to non-convex formulations because such terms break the positive semidefinite structure required for convexity. In our formulation, the multiplication of variables $w_i$ and $a_{i,j}$ directly results in these non-convex interactions.
> - For (P3): To restore convexity, we applied a reparameterization by defining new composite variables $\alpha_i^j=(1-p_i) \cdot w_i a_{i,j}$. With this transformation, the optimization variables become linearly coupled in the constraints, and the objective function reduces to a convex quadratic form in these new variables. Specifically, after reparameterization, the constraints are affine, and the objective becomes a convex quadratic function with positive semidefinite structure, clearly demonstrating convexity.
> \end{itemize}
>
> ***[Question-2]***
>
> In both proposed schemes, we set $\sum_{i=1}^k b_i = n + k - 1$, ensuring that each data partition is stored on at most two distinct workers. This construction ensures minimal redundancy.
>
> As a result, the average computation load (i.e., data replication factor) per partition is $d = 1 + \frac{k-1}{n}$, so each worker handles on average less than twice the amount of data compared to the ideal ($k \le n$). This replication factor is significantly lower than that of exact gradient coding schemes, which require at least $s+1$ data replicas per partition, where $s$ denotes the number of stragglers.
>
> By sharing only one partition between each worker pair, we achieve the necessary redundancy for straggler mitigation with the minimum overhead—reducing both per-node computation and memory requirements.
>
> ***[Suggestion-1 and Suggestion-2]***
>
> We thank the reviewer for these valuable suggestions to improve the paper's presentation.
> - Font Size: We will ensure the font size is consistent between the main body and the appendix in the revised version.
> - Equation (61): We agree that the step from Equation (60) to (62) is straightforward. Your suggestion is well-taken; we will remove the intermediate Equation (61) to make the proof in Appendix F.3 more concise and clear.
>
> Lastly, we would like to address the reviewer’s concern regarding the perceived simplicity of our approach. We respectfully contend that this simplicity is, in fact, a key strength—reflecting both elegance and practical utility. In contrast to prior works, which (i) were unable to accommodate heterogeneous straggler probabilities, limiting their applicability in real-world systems; (ii) relied on heavy data replication or binary coding schemes with inefficiencies; and (iii) focused solely on either residual-error minimization or unbiasedness of the gradient estimator, our work addresses all these limitations within a unified framework. Specifically, we are the first to explicitly formulate the optimal gradient coding problem under heterogeneous straggler settings, derive a universal optimal structure via Lagrangian analysis (Theorem 1), and propose two closed-form schemes (Schemes I and II) that operate with computation load $d < 2$.
>
> Our method ensures unbiased gradient estimation, near-minimal data replication, and provable convergence for both convex and non-convex objectives—all without requiring any runtime solver. Thus, the seemingly simple structure of our solution directly overcomes the key inefficiencies and limitations of prior approaches. We believe this integration of theoretical novelty and practical effectiveness constitutes a meaningful contribution to the field. Accordingly, we respectfully ask that the significance of these advancements be considered in the overall evaluation.

---

> ### Comment · Reviewer_dCZy · 2025-08-02
> **Further comments**
>
> **Concerns**:
>
> The authors' rebuttal addressed my concerns. In accordance with the policy in this year, the final rating (it will be made later) will not be disclosed to the authors.
>
> **Further questions**:
>
> For your Response-2 (or my Question-1), could you kindly provide the references (e.g., some textbooks) and the specific locations of the relevant content?
>
> Note that I think your explanation is reasonable. This is mainly for my personal learning purposes.
>
> **Further suggestions**:
> 1. Add the explanation for my Q2 in the revision. For example, this following sentence, or any other sentence you think is more appropriate.
> > By sharing only one partition between each worker pair, we achieve the necessary redundancy for straggler mitigation with the minimum overhead—reducing both per-node computation and memory requirements.
>
> 2. Writing in Section 3.2 (Reveiwer 4m6P's Weakness 2): In fact, this section also makes me a little confusing before I read this section very carefully. In my opinion, I think this can be due to the long text description, so I suggest adding some illustrative figures to show some representative examples for each scheme. For example, Assume there are 10 partitions. These figures show how Scheme I and Scheme II allocate these partitions.

---

> > ### Author Response · Authors · 2025-08-02
> >
> > Thank you once again for your constructive suggestions and your recognition of our contributions.
> >
> > ***[Further questions]***
> >
> > First, concerning the non-convexity of the bilinear term, we believe Section 3.1.4 ("Second-order conditions") and the example of a bilinear form in (3.59) of [r1] serve as helpful references. These illustrate how the coupling of two variables can undermine the problem's convexity.
> > Furthermore, regarding the transformation into a convex problem by substituting two variables with a single one, while this specific technique is not explicitly detailed under a single name in existing literature, we believe it is conceptually similar to established methods. It can be seen as a form of the convex transformation for hidden convexity as analyzed in [r2], or as bearing resemblance to the McCormick envelope [r3]. Depending on the problem's structure, Section 3.2 ("Operations that preserve convexity") of [r1] could also be relevant.
> > Additionally, for a discussion on the convex formulation and its solution properties, Section 5.5.3 ("KKT optimality conditions") of [r1] can be referred to for the underlying principles.
> >
> > ___References___
> >
> > [r1] S. Boyd and L. Vandenberghe, Convex optimization, Cambridge, U.K.: Cambridge Univ. Press, 2004.
> >
> > [r2] B.L. Gorissen, D. den Hertog, and M. Reusken, "Hidden convexity in a class of optimization problems with bilinear terms," Optimization Online, 2022.
> >
> > [r3] G. P. McCormick, “Computability of global solutions to factorable nonconvex programs: Part I—Convex underestimating problems,” Mathematical Programming, vol. 10, no. 1, pp. 147–175, 1976.
> >
> > ***[Further suggestions]***
> > - Regarding Q2: Thank you for the constructive suggestion. We will include the additional materials you requested in the revised version.
> > - Regarding the writing in Section 3.2: We appreciate this comment. In the revised version, we will add illustrative figures that clarify Schemes I and II, to improve readability and precision.

---

### Official Review · Reviewer_eHLb · 2025-07-02

**Clarity:** 3
**Significance:** 3
**Originality:** 3
**Rating:** 4
**Confidence:** 3

**Summary:**

This paper proposes an optimally structured gradient coding scheme to mitigate the straggler problem in distributed learning. To tackle with the stragglers in the real-world heterogeneous systems, an optimization problem is formulated in this paper to minimize the residual error while ensuring unbiased gradient estimation by explicitly considering individual straggler probabilities. Theoretical analysis of convergence is provided for strongly convex and smooth functions. The experiments show that the proposed algorithm can significantly mitigate the impact of stragglers and accelerates convergence.

**Questions:**

1. Is it possible to extend the theoretical analysis to non-convex or non-smooth functions?

2. Is the proposed algorithm also compatible to Adam optimizer?

**Ethical Concerns:**

["NO or VERY MINOR ethics concerns only"]

**Final Justification:**

The authors's feedback has addressed my concerns. The additional results on Adam looks good.
In overall, I don't see any major weakness.
Thus, I keep the positive score.

**Limitations:**

Yes, the authors adequately addressed the limitations and potential negative societal impact of their work

**Paper Formatting Concerns:**

No formatting issues are raised.

**Quality:**

3

**Strengths And Weaknesses:**

Strengths:

1. This paper proposes an optimally structured gradient coding scheme to mitigate the straggler problem in distributed learning. To tackle with the stragglers in the real-world heterogeneous systems, an optimization problem is formulated in this paper to minimize the residual error while ensuring unbiased gradient estimation by explicitly considering individual straggler probabilities.

2. Theoretical analysis of convergence is provided for strongly convex and smooth functions.

3. The experiments show that the proposed algorithm can significantly mitigate the impact of stragglers and accelerates convergence.

Weaknesses:

1. The theoretical analysis is limited to the strongly convex and smooth functions, which is less practical for real-world applications.

2. For the experiments, the optimizers are limited to SGD, which is not widely used these days compared to Adam.

---

> ### Author Rebuttal · Authors · 2025-07-30
>
> We sincerely thank the reviewer for their valuable feedback and insightful questions. We are encouraged that the reviewer found merit in our work. We address the specific weaknesses and questions below.
>
> ***[Weakness-1 and Question-1]***
>
> We thank the reviewer for their insightful comment regarding practical optimization settings.
> We would first like to clarify that our analysis already covers non-convex functions. Theorem 3 in our paper provides a formal convergence guarantee for general $\mu$-smooth, non-convex loss functions, ensuring our method converges to a stationary point.
> For the non-smooth case, while not explicitly detailed in the paper, our method's convergence is guaranteed under the well-established analysis in [Shamir et al.]. Specifically, the convergence analysis presented in [Shamir et al.] holds as long as two key assumptions are satisfied—both of which are met by our method:
> - Unbiased estimator: Our method is designed to provide an unbiased estimator of the gradient ($g^{(t)}$), as enforced by the condition in Equation (12).
> - Bounded variance: Our optimally structured coding scheme ensures a formal upper bound on the estimator's variance, as proven in Lemma 2.
>
> Since our algorithm meets these conditions, the convergence guarantees established by [Shamir et al.] for non-smooth settings can directly apply to our method. This confirms the theoretical robustness of our proposed method for a broad range of practical applications.
> We appreciate the opportunity to clarify this important theoretical connection and will add this discussion to the revised version of the paper.
>
> (O. Shamir and T. Zhang, ‘‘Stochastic gradient descent for non-smooth optimization: Convergence results and optimal averaging schemes,’’ in Proc. Int. Conf. Mach. Learn. (ICML), 2013, pp. 71–79.)
>
> ***[Weakness-2 and Question-2]***
>
> We appreciate the reviewer’s insightful comment and agree that evaluating our method with modern optimizers such as Adam is important. We would like to clarify that our framework is not restricted to GD/SGD; rather, it is compatible with any gradient-based optimizer. This is because optimizers like Adam and RMSProp first compute a gradient (or its estimator) and then apply their own update rules using this information.
> Our main contribution is an optimizer-agnostic approach for producing high-quality, unbiased gradient estimates in heterogeneous straggler environments. The optimization problem and resulting coding structure in Theorem 1 depend solely on straggler probabilities and are independent of how the gradient estimate is subsequently used. As a result, our method remains fully applicable when used with Adam or other gradient-based optimizers, while the achieved performance may vary depending on the optimizer used.

---

> > ### Comment · Reviewer_eHLb · 2025-08-02
> >
> > Thank you for the response.
> > The authors's feedback has addressed my concerns regarding to Weakness-1 and Question-1.
> > However, for Weakness-2 and Question-2, my concern is mostly about how the combination with Adam performs in practice and real-world applications. I do understand that the combination between the proposed algorithm and Adam is conceptually valid and feasible, but how it performs in real experiments is another thing. For example, there is a technique for gradient compression called error feedback mechanism, which also works well with SGD with strong theoretical analysis, but doesn't work very well with Adam-ish optimizers in practice.
> > That being said, I understand that the main contribution of this paper is the theory part.
> >
> > Thus, I would keep the current positive score.

---

> ### Author Response · Authors · 2025-08-08
>
> We sincerely thank the reviewer for their insightful feedback. The proposed method improves convergence by ensuring unbiasedness in the gradient estimator while reducing variance, which is effective for first-moment-based optimizers such as GD/SGD. However, for adaptive gradient methods that also estimate the squared gradient estimator, maintaining unbiasedness limits the reduction of the variance term. Specifically, preserving unbiasedness causes $E[\lVert g-\hat{g} \rVert^2_2]$ to accumulate as variance, which inflates the denominator in Adam’s update rule, shrinking step sizes and potentially degrading performance. This reflects a bias–variance tradeoff: forcing zero bias can hinder accurate squared gradient estimation.
>
> To address this, we extend the original gradient coding scheme into a two-track decoding framework: the original decoder is used to generate the gradient estimator for first-moment estimation, while a newly designed, slightly biased decoder (detailed below) is used to generate the gradient estimator for second-moment estimation. This enables us to preserve the unbiased estimation of the first moment, while introducing a controlled amount of bias in the second moment to reduce variance—ultimately improving both stability and accuracy. Note that the encoder remains unchanged, and the master node uses two decoders, introducing negligible overhead.
>
> The decoder for the second moment is obtained by solving:
> $\min_{v} \lambda [\sum_j (1-\sum_i (1-p_i)v_i a_{i,j})]^2 + [\sum_i p_i (1-p_i) v_i^2 (\sum_j a_{i,j})^2]$, where the first term corresponds to bias reduction, the second to variance reduction, and $\lambda$ determines the relative emphasis on reducing bias.
> Thus, after several derivations, we can obtain the closed-form solution $v_i^*=\frac{n \lambda}{p_i (\sum_{j=1}^n a_{i,j})(1+\lambda \sum_{m=1}^k \delta^{-1}_m)}$.
> Using Scheme I/II to generate the encoder $A$, we apply the original decoder $w$ for the first moment and the newly introduced decoder $v$ for the second, decoding each separately. Each decoder is used to decode the corresponding gradient estimator, from which the first- and second-moment terms are computed. These are then used to update the model following the update rule of optimizers such as Adam.
>
> In Figure 2(a) regeneration, Adam (centralized learning without stragglers) achieves 0.1071, Proposed (two decoders) 0.1092, Proposed (original, one decoder) 0.1440, SGC 0.1459, EHD 0.1420, BGC 0.1473, OD 0.1409, IS-SGD 0.1456. These results confirm the two-track decoder’s strong effectiveness and extendability to adaptive methods, with baseline strength largely due to Adam’s intrinsic noise suppression.
>
> This directly addresses the reviewer’s comments and demonstrates practical benefit. While more experiments and theory are planned, this work provides a foundational direction for extending variance-reduced decoding to adaptive optimizers.
>
> We would sincerely appreciate it if the reviewer could reconsider their rating, and potentially raise the score in recognition of the extended contributions presented above.

---

### Official Review · Reviewer_4m6P · 2025-07-03

**Clarity:** 2
**Significance:** 3
**Originality:** 2
**Rating:** 4
**Confidence:** 3

**Summary:**

This paper studies the structured gradient coding scheme to mitigate the straggler issue in distributed learning without relying on homogeneous straggler models or data replication. The authors derive closed-form solutions for the optimal coding and decoding coefficients and propose data allocation strategies to reduce the computation load.

**Questions:**

The questions are primarily included in the weaknesses part.

**Ethical Concerns:**

["NO or VERY MINOR ethics concerns only"]

**Final Justification:**

After the rebuttal, I would like to maintain my positive score of 4.

**Limitations:**

Yes

**Quality:**

3

**Strengths And Weaknesses:**

Strengths:
* The paper has a good motivation of improving the training efficiency in distributed learning, and gradient coding is important technique in distributed computing.
* The proposed method does not require too much additional data allocation.
* The theoretical analysis in section 3.1 is clear and easy to follow.

Weaknesses:
* We usually use (mini-batch) SGD for training in practice, and this paper also mentions batch sampling. However, it seems that the gradient $g^{(t)}$ is defined as the aggregated gradient at iteration $t$. It is unclear why the later analysis is not affected by the use of mini-batch gradients.
* The writing in section 3.2 could be further improved, I think the amount of data sharing and allocation is still a little bit confusing.
* The optimal structured gradient requires the known probability $p_1$ to $p_n$, which would be hard to predict during real application.
* The experimental setup is somewhat insufficient. The model used in the experiments is relatively small, especially in the context of the current LLM era. I would suggest that the authors include some scaling-up experiments to better demonstrate the practicality of the proposed method.
* Moreover, is the proposed method limited to gradient coding with SGD/GD optimizer? If so, it should be a limitation of the method for practical usage as well.

---

> ### Author Rebuttal · Authors · 2025-07-30
>
> We sincerely thank the reviewer for their insightful feedback. We appreciate the opportunity to clarify these important aspects of our work.
>
> ***[Weakness-1]***
>
> We thank the reviewer for their insightful question regarding mini-batch SGD. This prompted a deeper analysis that clarifies the trade-off between theoretical optimality and system efficiency, further highlighting the contribution of our work.
>
> Our method fundamentally builds upon and analyzes GD-like algorithms, but it can readily be extended to mini-batch SGD algorithms. In distributed learning, each distributed node $i$ can apply our proposed method by computing gradients using mini-batch sampling on their local data partitions ($\mathcal{D}_j, \forall j \in \mathcal{B}_i$). This approach can effectively reduce the update time per training iteration.
>
> However, batch sampling introduces randomness at each iteration. Hence, the expectation must now represent expectations over both the randomness from stragglers and mini-batch sampling at the $t$-th iteration, conditioned on the model parameter. Since the behaviors of stragglers and batch sampling are independent, the objective in problem (P3) can, after straightforward derivations, be reformulated as minimizing $\sum_{i \in [1:k]} \delta_i (\sigma^2 \sum_{j \in [1:n]} (\alpha_i^j)^2 + C (\sum_{j \in [1:n]} \alpha_i^j)^2)$ subject to the unbiasedness constraint $\sum_{i \in [1:k]} \alpha_i^j = 1$. Here, $\sigma^2$ denotes an upper bound on the variance of the mini-batch gradients, that is, $\mathbb{E}_\text{sample}[\lVert g^{(t)}_j - \bar{g}^{(t)}_j\rVert_2^2] \le \sigma^2$, where $\bar{g}^{(t)}_j$ is the  gradient computed from the mini-batch within data partition $\mathcal{D}_j$.
> The detailed derivations will be included in the revised version.
>
> This reformulated optimization problem, adapted to the mini-batch SGD setting, can be explicitly solved using the KKT conditions. This yields a dense, closed-form solution:
> $(\alpha_i^j)^* = \frac{\delta_i^{-1}}{\sum_{l \in [1:k]} \delta_l^{-1}}, \forall i \in [1:k], j \in [1:n]$.
> While this solution is theoretically optimal, it requires full data replication across all worker nodes ($d = k$), meaning that each node must store the entire dataset. This assumption is often impractical in distributed systems due to excessive storage and computation costs.
>
> On the other hand, our proposed schemes (Schemes I and II), which are formulated and constructed based on the GD algorithm, are specifically designed to maintain a low data replication factor, ensuring $d<2$ in both schemes.  When our proposed schemes, which are optimized for the full-batch GD setting, are directly applied to mini-batch SGD, the resulting gradient estimator does not achieve the theoretical minimum residual error with respect to the true gradient. Nonetheless, our approach maintains the important advantage of significantly reduced data replication under the mini-batch SGD setting.
>
> This observation highlights a trade-off when applying mini-batch SGD in distributed learning. Achieving the theoretically minimum residual error requires each worker node to handle a substantially increased computation load, which can accelerate convergence. In contrast, by utilizing our optimally constructed GD-based gradient codes and applying mini-batch sampling over the data partitions assigned to each worker, one may not reach the theoretical minimum of the residual error, but can substantially reduce the computation load per worker node.
>
> It is important to note that implementing mini-batch SGD using our proposed schemes results in a residual error bounded by $n^2 C \cdot  \frac{1}{\sum_{i=1}^k \delta_i^{-1}} + n \sigma^2 \bigg( 1+ \frac{n}{\sum_{i=1}^k \delta_i^{-1}}\bigg)$.
> This introduces an additional term, $n \sigma^2 (1 + \frac{n}{\sum_{i=1}^k \delta_i^{-1}})$, due to batch sampling, compared to the result in Lemma 2 of the paper; however, within our convergence analysis (Theorems 2, 3, and 4), the scaling term $n^2 C \cdot  ( 1+ \frac{1}{\sum_{i=1}^k \delta_i^{-1}})$ is only slightly modified to $n^2 C \cdot  ( 1+ \frac{1}{\sum_{i=1}^k \delta_i^{-1}} ) + n \sigma^2 (1+  \frac{n}{\sum_{i=1}^k \delta_i^{-1}})$ and thus the convergence rate remains essentially unchanged.
>
> ***[Weakness-2]***
>
> The goal of Section 3.2 is to present two specific data allocation and code design methods (Schemes I and  II) that satisfy the optimal conditions proven in Theorem 1, while also reducing the computation load caused by data redundancy in a practical distributed system.
> The key difference between the two schemes lies in their data sharing structure:
> - Scheme I: A single, specific data partition ($\mathcal{D}_1$) is a common partition shared by all workers, while the remaining partitions are assigned exclusively to individual workers. This scheme has a centralized sharing structure.
> - Scheme II: Each worker shares exactly one data partition with the worker of the adjacent index, so every partition is held by at most two workers. This scheme has a sequential and decentralized sharing structure.
>
> For further clarity, explicit construction examples of the matrix $\alpha$  for both schemes are provided in Appendix A.
> Both schemes are designed to reduce the computation load to $d=(n+k-1)/n$ by ensuring the total number of partitions held by each worker ($b_i$) sums to $\sum_{i \in [1:k]} b_i =n+k-1$.
> This is an efficient structure where only $k-1$ additional data replications are needed across the entire system, ensuring that each partition is computed $d$ times on average.
> Through these structural designs, we aimed to minimize the excessive data duplication and computational burden that can occur in real-world systems while maintaining theoretical optimality.
> Reflecting your feedback, we will improve our writing and add a new illustrative figure clearly demonstrating both sharing schemes with a concrete example, which should avoid any remaining confusion.
>
> ***[Weakness-3]***
>
> We acknowledged this as a limitation and  explicitly mentioned this issue in Appendix C.3 (Limitations) of our paper.
> Nonetheless, to overcome this limitation, a practical estimation approach can be applied in a real-world distributed learning environment:
> - The most practical approach is for the master node to estimate each worker's straggler probability by simply counting how often a worker exceeds the deadline, based on historical logs, and updating this estimate periodically. For example, the master can track how many times each worker was late out of the most recent tasks, and use this frequency as the current estimate of the straggler probability. This method is commonly known as empirical maximum likelihood estimation (MLE).
> - Another practical approach is to estimate each worker node’s straggling probability $p_i$ using a well-modeled estimator, as proposed in [18, 19]. Specifically, task completion times can be modeled using parametric probability distributions, such as the shifted-exponential distribution employed in [18]. By fitting this distribution to observed runtime data, we obtain reliable estimates of $p_i, \forall i$, representing the probability that worker $i$ fails to meet the deadline. These estimated values can then be directly integrated into our coding scheme as fixed parameters. Experimental results in [18], conducted on a real EC2 cluster, demonstrate the effectiveness of this method in enhancing the speed and robustness of distributed learning systems. In our own experiments, we adopted this same estimation approach for $p_i$ and based our evaluations on the resulting values.
>
> ***[Weakness-4]***
>
> Thank you for your valuable suggestions. Since the achieved performance depends on the specific model or dataset employed,  to address reviewer's concern, we have newly conducted  supplementary experiments using the state-of-the-art RetinaNet model to benchmark performance. RetinaNet, with approximately 34 million parameters, is roughly 6.3 times larger than the model (5.4 million parameters) originally employed in our paper. Due to restrictions on PDF uploads for this review, we  directly provide the  experimental results of each method. Specifically, we have reproduced the experiments depicted in Fig. 3 of the submitted paper with the new model. After 50 training iterations,
> for the threshold $\tau_{th}=1.1$, the loss values are as follows: GD achieves 0.1875, the proposed records 0.2178, SGC is 0.2485, EHD is 0.8201, BGC is 0.8840, OD is 0.8533, and IS-SGD reaches 1.1073.
> For $\tau_{th}=1.5$, GD remains at 0.1875, while the proposed achieves 0.2144, SGC is 0.2389, EHD is 0.2520, BGC is 0.2567, OD is 0.2839, and IS-SGD records 0.4120.
> Based on these results, it can be seen that the proposed method consistently maintains the performance trends identified in the paper, regardless of the model size.
>
> ***[Weakness-5]***
>
> We thank the reviewer for this insightful comment. We agree that evaluating our method with modern optimizers like Adam is important. We would like to clarify that our proposed framework is not limited to GD/SGD optimizer. This is because gradient descent-based optimizers—including GD, SGD, and Adam, and RMSProp—first compute a gradient or its estimator, and then update the model parameters according to their own specific update rules using this information.
> Our core contribution is an optimizer-agnostic method for generating a high-quality gradient estimate in heterogeneous straggler environments. The problem is formulated to find an unbiased estimator that minimizes the residual error (i.e., variance). This optimization and the resulting coding structure (Theorem 1) depend only on the straggler probabilities, not on how the gradient estimate is later used by an optimizer.  Therefore, the proposed method for calculating the estimator remains unchanged when using other gradient descent-based optimizers, while the achieved performance may vary depending on the choice of optimizer.

---

> > ### Comment · Reviewer_4m6P · 2025-08-05
> >
> > Thank you to the authors for the detailed response. I have no further concerns and will maintain my positive score.

---

> ### Author Response · Authors · 2025-08-08
>
> Thank you for carefully considering our response.
> Following the valuable suggestions from Reviewer 4m6P, we are pleased to provide an additional response regarding the extension of our method to adaptive gradient methods.
>
> Our proposed method improves convergence speed by ensuring unbiasedness in the gradient estimator while reducing variance. This has been effective for optimizers that use only the first moment, such as GD/SGD. However, in adaptive gradient methods—which also utilize the squared gradients for second-moment estimation—maintaining unbiasedness while reducing variance faces inherent limitations in decreasing the variance term of the second moment. The reason is that preserving unbiasedness inevitably causes $E[\lVert g-\hat{g} \rVert^2_2]$ to accumulate as variance. This, in turn, systematically inflates the denominator in Adam’s update rule, leading to excessive step-size shrinkage and possible performance degradation. In summary, this is a manifestation of the bias–variance tradeoff: forcing bias to zero can hinder squared gradient estimation, which inherently contains both bias and variance components.
>
> To address this, we additionally propose a two-track decoding approach. This scheme preserves the first-moment estimation of the original method while, for the second moment, allowing slight bias in exchange for reduced variance, thereby improving both bias and variance performance. The implementation introduces negligible overhead, as the encoding remains unchanged and the master node simply uses two decoding vectors during decoding.
>
> Since the encoder is fixed, we focus on designing the decoder. Designing a decoder that reduces both bias and variance leads to solving the following problem, where $v$ is used instead of $w$ for notational clarity:
> $\min_{v} \lambda [\sum_j (1-\sum_i (1-p_i)v_i a_{i,j})]^2 + [\sum_i p_i (1-p_i) v_i^2 (\sum_j a_{i,j})^2]$.
> The first term corresponds to bias reduction, the second to variance reduction, and $\lambda$ determines the relative emphasis on reducing bias. The solution can be derived by:
> $v_i^*=\frac{n \lambda}{p_i (\sum_{j=1}^n a_{i,j})(1+\lambda \sum_{m=1}^k \delta^{-1}_m)}$.
> Using Scheme I or II from the paper to generate the encoding matrix $A$, the master node applies the original $w$ for first-moment estimation and the above $v$ for second-moment estimation, decoding each separately to obtain the gradient estimators. These are then used to update the model following the update rule of optimizers such as Adam.
>
> Experimentally, when regenerating Figure 2(a), we obtained: Adam (centralized learning without stragglers) = 0.1071, Proposed (two decoders) = 0.1092, Proposed (original, one decoder) = 0.1440, SGC = 0.1459, EHD = 0.1420, BGC = 0.1473, OD = 0.1409, IS-SGD = 0.1456. These results confirm that the proposed two-track decoder yields significant performance improvements and shows extendability to adaptive gradient methods. The strong baseline performance primarily stems from Adam’s intrinsic noise suppression effect.
>
> In conclusion, this additional analysis addresses the reviewer’s comments and demonstrates the empirical effectiveness of the newly proposed two-track decoding scheme. Our study can be extended to adaptive gradient methods and serves as a foundational contribution that can be further validated through additional experiments and theoretical analysis.
>
> We would sincerely appreciate the reviewer’s reconsideration of the rating, and would be grateful if the score could be raised in light of the additional insights provided.

---

### Official Review · Reviewer_uuQF · 2025-07-04

**Clarity:** 4
**Significance:** 2
**Originality:** 2
**Rating:** 4
**Confidence:** 3

**Summary:**

The paper studies distributed gradient descent with the presence of straggler nodes. Linear coding is used to combat stragglers, by allocating segment $i$ of the data to $d_i$ workers, and sharing a linear combination of the gradients with a central server. The gradients are then (approximately) decoded (linearly) from the set of non-straggler nodes. The main technical problem the authors solve is: design the encoding and decoding matrices such that the gradient variance is minimized under the constraint that the gradient estimate is unbiased.
Under bounded gradient assumption, the optimization problem (variance and unbiasedness conditioin) is rewritten as a function of the encoding/decoding coefficients and formulated as a convex optimization problem.

**Questions:**

Could the authors discuss the worst-case computation load across nodes?

**Ethical Concerns:**

["NO or VERY MINOR ethics concerns only"]

**Final Justification:**

The authors have addressed my concerns. I choose to remain the original rating.

**Limitations:**

yes

**Quality:**

3

**Strengths And Weaknesses:**

Strength:
1. The paper is well-written and easy to follow.
2. The authors propose optimal (minimizing variance) encoding schemes that also have a low computation load (average number of replicates across partitions).
3. The condition of unbiased gradient is reasonable as it guarantees convergence of GD.
4. The authors additionally prove properties for the optimal encoding scheme.
5. Simulation results show better GD convergence compared to baselines.

Weakness:
1. The authors look for an encoding scheme with low computation load. However, the computation load is defined as the average number of repetitions for each partition, and does not take into account how these repetitions are allocated across nodes. Another important quantity is the load per worker node; namely: the maximum number of partitions allocated to a single worker.
2. While the original formulation of P2 is non-convex due to the multiplication between $w$'s and $a$'s. The products can be absorbed into a single variable to reach a convex optimization problem.

---

> ### Author Rebuttal · Authors · 2025-07-30
>
> We thank the reviewer for their thoughtful comments and questions. Below is our detailed response to address the concerns raised:
>
> ***[Weakness-1 and Question-1]***
>
> We agree that the maximum computation load on a single worker is a critical metric for practical deployment, and our framework is explicitly designed to address this. In our schemes, each worker $i$’s computation load is $b_i$, the number of assigned data partitions. Thus, the worst-case computation load is $\max_{i \in [1:k]} b_i$.
>
> Our method allows for flexible—though not entirely arbitrary—adjustment of the parameters $b_1, \ldots, b_k$ based on straggler probabilities and the computational capacity of each worker. The proposed Schemes I and II leverage the constraint $\sum_{i \in [1:k]} b_i = n+k-1$ to reduce the computation load, which inherently imposes a mathematical upper bound on the worst-case computation load, given by $\max_{i \in [1:k]} b_i \leq n-k+2$. For instance, this bound can be achieved by assigning $b_1 = n-k+2$, $b_2 = 2, \ldots, b_{k-1} = 2$, and $b_k = 1$. Moreover, assuming the total dataset can be partitioned into subsets of approximately equal size, the number of data partitions $n$ can also be selected accordingly. Under this scenario, choosing $n = k$ ensures that the worst-case computation load is at most $2$.
>
> Thus, system designers can flexibly constrain $\max_{i \in [1:k]} b_i$ based on specific hardware limitations, effectively preventing any single worker from becoming overloaded while still optimizing overall system performance. For example, Appendix A presents a scenario with $n = 4$ data partitions and $k = 3$ worker nodes, where the allocation parameters are set as $b_1 = 3$, $b_2 = 2$, and $b_3 = 1$. In this case, the worst-case computation load is $\max_{i \in [1:k]} b_i = 3$. To reduce this worst-case load, the parameters can be rebalanced to $b_1 = 2$, $b_2 = 2$, and $b_3 = 2$, resulting in a lower worst-case computation load of $\max_{i \in [1:k]} b_i = 2$.
>
> Importantly, the average computation load $d = (n + k - 1)/n$ remains strictly below 2 (since $k \leq n$), ensuring overall efficiency. In summary, our framework offers both per-worker load control and high average efficiency, making it well-suited to heterogeneous real-world systems.
>
> ***[Weakness-2]***
>
> The reviewer is correct in observing that the non-convex problem (P2) can be made convex via variable substitution. This is precisely the procedure we present in our paper to motivate our solution.
> Our narrative was structured to show the logical progression from the initial problem formulation to a solvable one.
> - We first formulate the problem (P2) and explicitly identify it as non-convex due to the strong coupling between the encoding and decoding variables.
> - We then introduce the variable substitution $\alpha_i^j=(1-p_i) \cdot w_i a_{i,j}$ stating that (P2) can, without loss of optimality, be transformed into (P3), which is a convex problem.
>
> Our core contribution is not the transformation itself, but the steps that follow:
> - Deriving the optimal structure (Theorem 1) that characterizes any solution to this convex problem.
> - Using this structure to develop two novel, closed-form code construction schemes (Scheme I and II) that are computationally efficient, bypass the need for a runtime convex solver, and have a low computation load.
>
> We thank the reviewer for allowing us to emphasize that the convex transformation is a key methodological step that enables our main contributions.

---

> > ### Comment · Reviewer_uuQF · 2025-08-08
> >
> > Thanks to the authors for addressing all my concerns.

---

> ### Author Response · Authors · 2025-08-08
>
> Thank you once again for your constructive feedback and for acknowledging the value of our contributions.

---

### Note · Authors · 2025-08-12

We once again thank all reviewers and the AC for devoting their time and thoughtful efforts to reviewing our paper. Below we concisely recap what changed or was clarified, with concrete follow-ups.

***(1) Per-worker worst-case load is bounded and tunable.***

We clarified that our schemes enforce $\max_i b_i \le n-k+2$ under $\sum_i b_i=n+k-1$, so designers can cap the heaviest worker while keeping average load $d=(n+k-1)/n<2$. We provided concrete rebalancing examples that lower the worst-case load without harming overall efficiency.

***(2) Beyond GD: mini-batch SGD and non-smooth cases.***

We formalized expectations over both stragglers and batch sampling, preserving unbiasedness and the structure of our bounds. For non-smooth objectives, our estimator maintains the standard assumptions (unbiasedness, bounded second moment), so established SGD-style convergence guarantees apply; we will include the precise statement and pointers.

***(3) Scaling experiment.***

We reported larger-scale results (e.g., RetinaNet, ~34M parameters) showing consistent trends versus baselines under higher model complexity.

***(4) Exposition improvements committed.***

We will include schematic figures for Schemes I and II—with a worked example in Appendix A—and revise Section 3.2 to improve clarity.

***(5) Optimizer-agnostic core + Adam optimizer extension.***

Our coding/decoding pipeline is optimizer-agnostic because the gradient estimator is formed before applying any update rule. To address Adam’s sensitivity to second-moment variance, we additionally introduce a two-track decoder: (i) retain the original unbiased first-moment estimator for stability of the update direction, and (ii) use a lightly biased decoder for the second moment to reduce its variance. This yields improved stability in Adam with negligible overhead (encoder unchanged; two decoding vectors at the master). We will include the formal derivation and implementation details.

Through our rigorous rebuttal, we were able to resolve most reviewers’ concerns. We believe these contributions collectively provide both theoretical novelty and practical value for the systems and infrastructure community.

---

### Decision · Program_Chairs · 2025-09-17

**Decision:**

Accept (poster)

**Comment:**

This paper proposes an optimally structured gradient coding scheme to mitigate the straggler problem in distributed learning. To tackle this in real-world heterogeneous systems, an optimization problem is formulated to minimize the residual error while ensuring unbiased gradient estimation by explicitly considering individual straggler probabilities. Theoretical analysis of convergence is provided for strongly convex and smooth functions. The experiments show that the proposed algorithm can significantly mitigate the impact of stragglers and accelerates convergence.

The reviewers noted the practical applicability of solving the ‘straggler problem’ in distributed learning, and therefore faster learning overall. The reviewers appreciated the clarity of the work, and the theoretically proven optimality of the encoding under the statistical assumptions (bounded gradient).  The optimization (min-variance and unbiased estimation) is written as a function of the encoding/decoding coefficients and convexity is shown, and the optimizer is relatively low complexity.  The primary contribution of the paper was considered to be the theoretical analysis and derivation of an optimal approach with guarantees.

There was considerable author-reviewer interaction and in final scoring all reviewers regarded the work positively.

The reviewers raised some secondary questions regarding applicability beyond SGD, such as when applying mini-batch and/or Adam.  They also wondered about robustness for non-smooth cases, and applicability for larger size model problems.  These are good areas for potential follow on research by the authors.